# Mountain permafrost degradation documented through a network of permanent electrical resistivity tomography sites

Coline Mollaret[1], Christin Hilbich[1], Cécile Pellet[1], Adrian Flores-Orozco[2], Reynald Delaloye[1], Christian Hauck[1]

[1]Department of Geosciences, University of Fribourg, Switzerland
[2]Department of Geodesy and Geoinformation, TU Wien, Austria

*Correspondence to*: Coline Mollaret (coline.mollaret@unifr.ch)

**Abstract.** Mountain permafrost is sensitive to climate change and is expected to gradually degrade in response to the ongoing atmospheric warming trend. Long-term monitoring of the permafrost thermal state is a key task, but problematic where temperatures are close to 0°C, because the energy exchange is then dominantly related to latent heat effects associated with phase change (ice/water), rather than ground warming or cooling. Consequently, it is difficult to detect significant spatio-temporal variations of ground properties (e.g. ice-water ratio) that occur during the freezing/thawing process with point scale temperature monitoring alone. Hence, electrical methods have become popular in permafrost investigations as the resistivities of ice and water differ by several orders of magnitude, theoretically allowing a clear distinction between frozen and unfrozen ground. In this study we present an assessment of mountain permafrost evolution using long-term electrical resistivity tomography monitoring (ERTM) from a network of permanent sites in the Central Alps. The time series consist of more than 1000 datasets from six sites, where resistivities have been measured on a regular basis for up to twenty years. We identify systematic sources of error and apply automatic filtering procedures during data processing. In order to constrain the interpretation of the results, we analyse inversion results and long-term resistivity changes in comparison with existing borehole temperature time series. Our results show that the resistivity data set provides valuable insights at the melting point, where temperature changes stagnate due to latent heat effects. The longest time series (19 years) demonstrates a prominent permafrost degradation trend, but degradation is also detectable in shorter time series (about a decade) at most sites. In spite of the wide range of morphological, climatological, and geological differences between the sites, the observed inter-annual resistivity changes and long-term tendencies are similar for all sites of the network.

## 1    Introduction

Mountain permafrost is sensitive to climate change and has been affected by a significant warming trend in the European Alps for the last two decades (Noetzli et al., 2016). The warming trend is however not uniformly distributed (both spatially and

temporally), as permafrost occurs in a large variety of complex landform settings. To understand the site-specific and regional evolution of mountain permafrost, detailed information about the spatio-temporal permafrost distribution and its monitoring is necessary. A better understanding of freeze and thaw processes is critical as thawing permafrost will affect mountain slope stability and trigger rock falls (Ravanel et al., 2017).

5 The thermal definition of permafrost (ground remaining at or below 0 °C during at least two consecutive years) led the scientific research into ground temperature monitoring, both at regional - such as the well-established borehole networks in Switzerland (PERMOS, 2019) and Norway (NORPERM, Juliussen et al., 2010; Isaksen et al., 2011) - and global scales (within the Global Terrestrial Network of Permafrost, GTNP, Biskaborn et al., 2015). However, the permafrost definition does not necessarily imply the presence of ground ice or even any freezing and thawing processes: saline permafrost or dry permafrost do not 10 necessarily contain any ice and the 0 °C limit does not take a freezing point depression into account (Harris et al., 1988).

Ground temperature monitoring alone cannot identify freezing and thawing processes, which are of particular interest. At the freezing point, the effect of latent heat flux maintaining temperatures near 0 °C prevents the use of temperature measurements to analyse freezing and thawing processes. Furthermore, temperatures give only partial information regarding water content, and whether it occurs in liquid or solid phase. Unfrozen water content can indeed be high even at negative temperatures 15 especially for fine-grained soils. Ice content can be quantitatively estimated from direct drill core analysis (Haeberli et al., 1988), from nuclear well logging (Scapozza et al., 2015) and from combined geophysical surveys (Hauck et al., 2011; Hausmann et al., 2007; Maurer and Hauck, 2007; Pellet et al., 2016; Vonder Mühll and Klingelé, 1994).

Geophysical methods and especially geoelectrical techniques can accurately distinguish between frozen and unfrozen soil, as the electrical resistivity is particularly sensitive to the presence of liquid water and temperature permitting an indirect 20 quantification of temperature changes in the subsurface with higher spatial resolution than point-scale temperature loggers (e.g. Hauck and Kneisel, 2008a). The electrical resistivity (or its reciprocal, the electrical conductivity) combines the contribution of three conduction mechanisms: particle conduction, surface conduction, and electrolytic conduction (e.g. Klein and Carlos Santamarina, 2003). Particle or electronic conduction is mainly related to metallic materials (due to the high number of free electrons). Surface conduction mechanisms take place at the electrical double layer formed at the mineral-ice-water 25 interface (at the electrical double layer) and is controlled by the water content and the electrical properties of the interface (pH, zeta-potential, cation exchange capacity). Finally, the ionic or electrolytic conduction takes place through fluid-filled pores and is controlled by porosity, saturation, pore connectivity, and fluid conductivity. The flow of electrical current in soils/rocks is mainly controlled by ionic charge transport taking place through water-filled pores, and by surface conduction mechanisms at mineral-ice-water interfaces (especially in the case of clay-rich materials, e.g. Revil and Glover, 1998). Hence, spatial and 30 temporal resistivity variations in the subsurface depend up on porosity, pore connectivity, mineralogy, water chemistry, water content, water saturation and temperature (e.g. Duvillard et al., 2018; Hermans et al., 2014; Ward et al., 2010). Upon freezing, electrolytic conduction paths are severely reduced, reducing ion mobility and implying an increase of resistivity of (partially) frozen soil/rock, which can be monitored if the contact between electrodes and the ground (the so-called contact resistance)

permits current injection and measurement of voltages (Hauck, 2002). According to Duvillard et al. (2018), the surface conduction dominates at low salinity instead of the electrolytic conduction.

Geoelectrical methods therefore permit detection and monitoring of both ice melting (through the phase change from ice to water) and thawing (through the higher mobility of ions for higher temperatures) processes (Hauck, 2002; Oldenborger and LeBlanc, 2018). The considerable resistivity change during freezing and thawing processes is (in addition to the temperature dependence of electrical resistivity, Hayley et al., 2007; Oldenborger and LeBlanc, 2018) mainly a consequence of the inverse conductivity characteristics of ice (characterized by high electrical resistivity values) and water (characterized by high electrical conductivity values). Geoelectric surveying adds a spatial dimension and it can detect changes occurring at temperatures close to the freezing point during the so-called zero curtain period (defined in 1990 by Outcalt et al.) when temperature does not vary significantly.

Electrical resisitivity tomography (ERT) and its monitoring variant ERTM has been used for many years in other geoscientific fields such as landslide investigations, irrigation/infiltration studies, landfill monitoring and remediation of contaminated sites (e.g. Barker and Moore, 1998; LaBrecque and Yang, 2001; Loke, 1999; Suzuki and Higashi, 2001; Flores Orozco et al., 2019). However, although ERT is nowadays recognized as a standard method in permafrost surveying (e.g. Kneisel et al., 2008), long-term ERTM profiles have only relatively recently been introduced in this research domain (Doetsch et al., 2015; Farzamian et al., 2019; Hilbich et al., 2008a, 2008b; Keuschnig et al., 2017; Kneisel et al., 2008, 2014; Lewkowicz et al., 2011; Oldenborger and LeBlanc, 2018; Supper et al., 2014; Tomaškovičová, 2018).

In a recent review of published permafrost ERTM approaches, Supper et al. (2014) presented the acquisition parameters of eight sites, of which two are part of the ERTM network presented in this study. Since then, further ERTM case studies on permafrost were published that focused, for example, on rock wall monitoring (Keuschnig et al., 2017), combining direct-current resistivity with induced polarization monitoring (Doetsch et al., 2015), or analysing vegetation-permafrost interaction (e.g. Dafflon et al., 2017; Stiegler et al., 2014). However, in contrast to borehole temperature monitoring networks, most of the above-cited ERTM studies focused on single field sites or on short-term monitoring (< 5 years).

Mountain permafrost terrain is indeed challenging regarding the installation and maintenance of long-term (> 5 years) monitoring stations, requiring a high effort due to remoteness, field instability, harsh weather conditions (specially the risk of lighting damage to the instrument, as described by Supper et al., 2014) and high ground resistance (Hauck, 2013). Maintenance of these sites also requires a long-term commitment of geophysically trained personnel, which keeps the number of existing monitoring sites low. Here, we present an established ERTM network of six alpine permafrost sites, the corresponding procedures for automated and comparable data processing, and the results of complementary borehole temperature and resistivity time series of more than a decade. We demonstrate the benefit of joint ERTM data processing and interpretation within a regional network as complementary 2-dimensional information about freezing and thawing processes in mountain permafrost compared to point-scale borehole data. The purpose is to assess permafrost degradation over one to two decades, investigate the correlation between temperature increase and resistivity decrease, and explore differences between geologically and climatically different alpine sites.

## 2 Field sites and data set

### 2.1 ERTM network

ERTM in Switzerland started in 1999 at Schilthorn/Berner Oberland (Hauck, 2002), as part of the European PACE project (Permafrost and Climate in Europe, Harris et al., 2009). Due to its promising results, the installation was maintained and later

extended to a fully automated and continuously operating monitoring site (Hilbich et al., 2011). During the following decade, ERTM was initiated at an increasing number of sites; four of them were subsequently included in the Swiss permafrost monitoring network PERMOS (Hilbich et al., 2008a; Morard, 2011; PERMOS, 2019). Though other ERTM sites exist in the Central Alps, the network presented here is composed of a selection of six monitoring sites located in different regions of the Swiss and Italian Alps located in Fig. 1. These six sites (out of the 14 sites, where repeated measurements are available, Table

1) were selected according to a) length of ERTM time series; b) existence of long borehole temperature time series for validation and process analysis; and c) diversity of subsurface properties. Table 1 summarizes the general characteristics of the ERTM network in addition to other ERTM sites introduced below. In this paper, we present datasets until the end of 2017. The network covers a large variety of site characteristics regarding substrate, ice content, geomorphology, thermal properties and the observed soil temperature range. The active layer thickness (ALT, i.e. the maximum seasonal thaw layer that is here

derived from temperature measurements) given in Fig. 2 varies for all sites between 2.5 and almost 10 m, depending on site characteristics and local climatic conditions (PERMOS, 2019; Pogliotti et al., 2015). During the observation period, a general increase in ground temperatures was observed in all boreholes (PERMOS, 2016; Pogliotti et al., 2015). Figure 2 summarises the main thermal characteristics of all sites (Mean annual air temperatures (MAAT), ground temperature at 10 m depth, and ALT) compared to the average resistivity at around 10 m depth ($\rho_{10m}$) at the borehole location. The available ERTM time

series of the respective sites shown in Fig. 3 highlight the almost 20-year time series at Schilthorn (SCH) and the ~ 10 year period at most other sites of the network. Figure 1 and Table 1 show additional existing ERTM time series from permanent electrode arrays exist, but they are not the focus of the present paper due to a) the absence of borehole information or b) interruption of measurements for safety reasons.

### 2.2 Field site descriptions

#### 2.2.1 Schilthorn (SCH)

The Schilthorn monitoring site is located in the Bernese Oberland, Northern Swiss Alps (46°33′30″N 7°50′05″E) at a height of 2910 m above sea level (asl) on the north-facing slope of the summit. The lithology consists of ferruginous sandstone schists that have weathered to produce a fine-grained surface debris layer up to several metres thick (Hilbich et al., 2008b; Vonder Mühll et al., 2000). The permafrost presence at the Schilthorn massif was first reported during the construction of the cable

car station in 1965 (Vonder Mühll et al., 2000).

The monitoring station comprises a meteorological station, three instrumented boreholes, three soil moisture sensors and ERT profiles. Soil moisture has been monitored since August 2007 from 10 to 50 cm depth (Hilbich et al., 2011; Pellet et al., 2016).

The meteorological station includes sensors measuring shortwave solar and longwave atmospheric and terrestrial radiation, air temperature, humidity, snow height, wind speed and direction. Three boreholes, including two along the ERTM profile, (14-m deep SCH-5198 drilled 1998) and 100-m deep SCH-5000 drilled 2000) are instrumented with temperature loggers (cf. PERMOS, 2016; Vonder Mühll et al., 2000 for details of the borehole instrumentation). Permafrost depth extends to at least 100 m and ALT varies between 5 and 10 m (PERMOS, 2016). The ERTM profile at SCH is horizontal and almost parallel to the Schilthorn east-west striking crest.

### 2.2.2 Stockhorn (STO)

The Stockhorn monitoring site is located on a high-altitude plateau surrounded by glaciers above Zermatt in the canton of Valais (45°59′12″N 7°49′27″E at 3410 m asl) on the ridge between Gornergrat and Stockhorn summits. Composed of an alternation of micaschist and gneiss, the near-surface weathered bedrock is covered by a surface layer of coarse debris with a thickness ranging from a few centimetres to a metre. The plateau separates a steep glacier-covered north face from a more gently inclined slope to the South, where bedrock outcrops are visible (Gruber et al., 2004).

The site comprises a full meteorological station, continuous soil moisture measurements, and a permanent ERTM profile (Hilbich et al., 2008a; Pellet and Hauck, 2017; PERMOS, 2019; Rosset et al., 2013). Figure 2 shows the considerable spatial variability of permafrost temperatures and active layer depth at the two boreholes. After initial geophysical measurements in 1998 during the PACE project (Hauck, 2001), a permanent ERTM profile was installed in 2005, extending from the relatively flat plateau to the south slope. In 2007, the profile was extended with seven additional electrodes drilled into the steep north face to get a more convex geometry (cf. Table 1).

### 2.2.3 Lapires (LAH)

The Lapires monitoring site is a high alpine talus slope located in the Nendaz valley, canton of Valais (46°06′22″N 7°17′04″E) at a height of 2500 m asl. Ground ice was discovered in 1998 during the construction of cable car pylons on the north-facing slope of the Col de Chassoure (Delaloye and Lambiel, 2005). In addition to its high-alpine location, this talus slope is characterized by the occurrence of internal air circulation, with ascent of warm air in winter and descent of cold air in summer (Wakonigg, 1996; Delaloye and Lambiel, 2005; Wicky and Hauck, 2017). This so-called chimney effect causes a net cooling of the scree slope compared to surrounding areas at the same altitude without ventilation, which may under favourable conditions even lead to the development of low-altitude permafrost.

Ground temperatures have been monitored since 1999 (borehole LAP-0198, 19.6 m depth; PERMOS, 2016), and in three additional boreholes (up to 40 m depth) since 2008 (Scapozza et al., 2015), two of which are in permafrost (LAP-1108 and LAP-1208; PERMOS, 2016). Using nuclear well logging at these boreholes, Scapozza et al. (2015) quantitatively estimated the porosity (40-80 %) and the volumetric ice content (20-60 %) from 4 to 20 m depths. A meteorological station completes the temperature dataset (Staub et al., 2015). ERTM started in 2006 with a permanent horizontal profile (LAH, cf.Table 1),

complemented by an additional, longer vertical cross-profile (LAV) in 2007. The LAV profile (not part of this paper) was damaged due to frequent rockfall and was finally uninstalled in 2017.

### 2.2.4  Murtèl-Corvatsch (MCO)

The Murtèl-Corvatsch monitoring site is a coarse blocky crystalline rock glacier located on a northern slope below Piz Corvatsch in Upper Engadin, Eastern Swiss Alps (46°25′44″N 9°49′18″E) at a height of 2670 m asl and contains the longest borehole temperature record (since 1987) in the European Alps. Drill core analysis revealed an average ice content of 80-100 % (Haeberli et al., 1998). The rock glacier is characterized by well-defined transversal ridges and furrows on its lower part (Fig. 1). Inclinometer data in the borehole locate a shear zone at ~ 30 m depth (Arenson et al., 2002), while bedrock was encountered at 57 m depth (from borehole data and from an additional ERT survey with 10 m spacing, not shown). Further boreholes drilled in 2000 and 2015 (PERMOS, 2016) detected significant intra-permafrost water flow (Arenson et al., 2010). Within the different boreholes, ALT varies between 2.5 and 3.5 m (PERMOS, 2016).

Similar to SCH and STO, a full meteorological station data are available since 1997. First ERT data were acquired in 1998 as part of the PACE project along a longitudinal profile (Vonder Mühll et al., 2000), and at approximately the same position a long-term ERTM profile was installed in 2005.

### 2.2.5  Cervinia (CER)

The Cervinia monitoring site is located on a high altitude plateau at Cime Bianche in the upper part of Valtournenche (45°55′09″N 7°41′34″E) at a height of 3100 m asl in the Northern Italian Alps (Pogliotti et al., 2015). The bedrock surface is highly weathered resulting in a surface layer of fine-grained to coarse-grained debris of micaschists (up to a couple of metres thick). First indications for potential permafrost occurrence at the site were derived from bottom temperature of snow (BTS) measurements and the presence of gelifluction lobes (Pogliotti et al., 2015). Ground temperature is monitored in two boreholes of 7 and 41 m depth, which were drilled in 2004 by ARPA (Environmental Protection Agency of Aosta Valley). Temperatures and ALT (see Fig. 2) exhibit a pronounced spatial heterogeneity due to differences in ice/water content and bedrock weathering degrees (Pogliotti et al., 2015). An automatic meteorological station including soil moisture sensors was installed in 2006. ERTM started in 2013 and data collection is repeated at least once a year (Pellet et al., 2016) in collaboration with ARPA.

### 2.2.6  Dreveneuse (DRE)

The Dreveneuse monitoring site is a low altitude and vegetated northwestern scree slope (Morard, 2011) located in the so-called Prealps, Canton of Valais (46°16′24″N 6°53′22″E) at a height of 1580 m asl. This scree slope consists of loose limestone debris, and it is affected by a reversible internal air circulation process (similar to LAH). Two boreholes were drilled and instrumented in 2004 in this scree slope, and one of the boreholes (14.5 m deep) exhibited year-round negative temperatures as a result of a few relatively cold winters. This "short-term permafrost" was only observed in 2005-2006 and 2010-2011 (Morard, 2011). Since then the borehole only exhibits seasonal freezing. Due to its spatially heterogeneous internal air

circulation and the feedback with vegetation and moisture, DRE is however considered an important location at the permafrost limit (Morard et al., 2008, 2010; Pellet and Hauck, 2017), and is used in this investigation for comparison with ERTM data from the other network sites.

The site is further equipped with a meteorological station including soil moisture measurements (Pellet and Hauck, 2017). The permanent ERTM profile was installed in 2007. Initially, electrical resistivity measurements were conducted on a monthly basis (Morard, 2011), however since 2012, ERTM measurements occur about once a year.

## 3    Methods

### 3.1    Data acquisition

ERT is used to image the distribution of the electrical resistivity of the subsurface, based on a 4-electrode array, or quadrupole. The electrical voltage (or potential difference) $V$ is measured between a pair of electrodes while a current at intensity $I$ is injected between a second pair of electrodes. Tomographic measurements deploy tens of electrodes for the collection of a few hundreds of quadrupoles, which in combination with adequate inversion algorithms permits a solution for the distribution of the electrical resistivity of the subsurface (e.g. Dahlin, 2001; LaBrecque et al., 1996).

Long-term ERTM requires consistency in all acquisition parameters for comparison purposes. Regarding the configuration of the quadrupoles, surveys were collected in early years using one of three standard configurations: Wenner (W), Wenner-Schlumberger (WS) and Dipole-dipole (DD) (e.g. Hauck et al., 2003). Recently, the W configuration has been established as the main configuration used for the monitoring measurements, because of a high signal-to-noise-ratio. Even though logistical constraints have changed at some sites and new instrumentation became available since the earliest measurements, the same W configuration is still deployed to permit a fair comparison of the data from different periods. However, CER and DRE are exceptions as their monitoring procedure continues to use the initial WS configuration.

We use two different instruments for the acquisition of ERTM data: a SyscalR1 instrument (Iris Instruments, France) with an internal resistivity of $10^8$ $\Omega$, which can connect to 48 electrodes, and a Geotom instrument (Geolog, Germany) with an internal resistivity of $10^9$ $\Omega$, which can connect to 50 electrodes. Permanently installed stainless steel electrodes are used for current injection and potential measurements. Contact resistances were measured before every collection, which high values related to a poor contact between electrode and ground play, typical from high resistive media like alpine geology and frozen ground. High contact resistances limit the current injections and have a negative effect in the data quality. Thus, different strategies to deal with high contact resistances in permafrost investigations have been addressed in several publications (e.g. Hauck and Kneisel, 2008a; Ingeman-Nielsen et al., 2016; Tomaškovičová et al., 2016). For all our ERTM sites, we tried to maximize the contact surfaces between electrode and ground by either pouring fresh water onto the ground or connecting several electrodes in parallel to reduce contact resistances. Electrode separation varies between 2 and 5 metres (see Table 1) accounting for the characteristics of each site and the envisaged depth of investigations, which is between 15 and 40 m for the different sites.

## 3.2 Pre-inversion data processing

In the context of long-term and automated ERTM at several sites, a consistent processing of the data prior to the inversion is required to evaluate data quality. This has already been addressed for single automated stations in mountain permafrost terrain

(Hilbich et al., 2011; Keuschnig et al., 2017; Supper et al., 2014). In a precursor study, a data filtering scheme for several ERTM data sets for the Swiss Alps has been developed (Rosset et al., 2013). Here, we extend this approach by analysing the seasonal and site-specific characteristics of contact resistance and the statistical distribution of measured apparent resistivities. By this, we identify and remove outliers before the inversion of the data to reduce possible artefacts.

### 3.2.1 Electrode contact resistance

Galvanic contact resistances between electrodes and ground are recorded before each measurement at SCH, LAH, MCO, and STO, where the Geotom system is deployed. The recorded values are below the internal instrument resistance except for singular cases, where the filtering scheme identified them accordingly. As observed in Fig. 4, electrode contact resistance exhibits site-specific characteristics and seasonal variations. Within one profile, the contact resistance characteristics of individual electrodes show consistency in time, reflecting substrate characteristics (e.g. in Fig. 4c the first six electrodes placed

in the steep north face always exhibit a higher resistance than most other electrodes of the profile). Similarly, the seasonal evolution of contact resistances is consistent between different sites and years, with low values in summer and high values in winter. The variations of contact resistance are most pronounced during seasonal surface freeze-up and melt (see e.g. mid-October in 2015 and mid-May in 2016 at LAH in Fig. 4e).

The median contact resistances exhibit a clear summer-winter duality at all sites (Figs. 4b, 4d and 4f). The lowest contact

resistances are observed at SCH (moist and fine-grained material) with median values varying from 1.5 kΩ in summer to 4.5 kΩ in winter. The median values recorded at LAH (coarse blocks, high temperatures) span from 20 kΩ in summer to 190 kΩ in winter. STO (medium blocks at the surface, low temperatures) exhibits higher median contact resistances from 68 kΩ in summer to more than 1000 kΩ in winter. Contact resistances are highest at MCO (due to coarse blocks at the surface) with already high median values of 120 kΩ in summer and insufficient contact in winter (not recorded). At MCO, winter

measurements were attempted (nine existing datasets) in 2007 and 2008 at different periods from December to May. However, the high standard deviation of the data led to lower data quality and datasets were consequently not further processed (Hilbich et al., 2009).

A similar analysis can be performed with injected current (see the distribution of injected current in Fig. S1 in the Supplement). Narrower current distribution results from more homogeneous surface conditions, while low current injections are associated

with high contact resistances, more outliers and often limited data reliability.

### 3.2.2 Filtering process

A consistent data processing is required for a comparative analysis of data collected in different locations and under variable conditions. In relatively resistive terrain, anomalously high resistivity values can indicate ground ice occurrences and freeze or thaw processes, but such values could also be associated to systematic errors. Such systematic errors can relate to low current densities injected (associated with high contact resistances) below the resolution limit of the ERT instruments. In view of the huge amount of data sets, automatic filtering procedures are essential to ensure consistent data processing. Several approaches with different levels of complexity exist. A time-lapse inversion scheme with time regularization (see e.g. Karaoulis et al., 2013; Lesparre et al., 2017; and the discussion section below) should in principle filter out the data, which are not correlated over time. The comparison of normal and reciprocal measurements (e.g. Flores-Orozco et al., 2012, 2018) is another common method to quantify data uncertainty. Reciprocals refer to the re-collection of the data after interchanging the electrodes used for current and potential dipoles. However, the collection of reciprocals also requires doubling the time necessary for the collection of the data, which is not well suited for monitoring measurements within the network. Except for singular tests, no reciprocal measurements to quantitatively estimate measurement noise levels were conducted, due to energy consumption and time constraints. In the absence of reciprocal measurements and 4D inversion schemes, we use a three-step procedure. The first step filters single quadrupoles, while the second and third steps reject entire data sets if the resolution or representativeness of the obtained specific resistivity model with respect to the measured values (raw data) is considered too low.

Step 1 includes a systematic filter (SF) and a moving median filter (MMF). This analysis is performed for a complete tomographic data set and filters individual quadrupoles. Following Hilbich et al. (2011), we apply a SF to remove physically implausible measurements (negative potential differences or apparent resistivity) and low-reliable measurements (i.e. low current flow associated with high potentials ($I < 0.01$ mA and $V > 100$ mV)). A MMF is then applied to remove outliers within individual depth levels of the data set (cf. Rosset et al., 2013). The length of the moving window was defined empirically. A quadrupole measurement is deleted from the data set if its deviation from the median within the moving window is higher than a pre-defined threshold (see Table 1). The MMF is applied twice (iteratively, with different window lengths and threshold values). The performance of the MMF is however limited by the data quality. Indeed, this technique detects spatial outliers among "good or normal" data (an example is given in the Supplement, see Fig. S2). If the ratio of "bad or outlier" data over "good" data is too high, the MMF technique will not yield acceptable results. In this case, however, the data set will very probably not fulfil the remaining filter criteria and will be filtered out during steps 2 or 3.

Step 2 deletes tomographic data sets, from which more than 20 % of the quadrupoles were removed in step 1. Rosset et al. (2013) defined a "cut-off" limit of 80 % for acceptable data sets by comparing the ratio of filtered data points with the subsequent inversion error. Only data sets containing more than 80 % of the original data points after filter step 1 are therefore inverted. Figure 5 shows the distribution of the percentage of data kept after filter step 1 for each data set as a function of seasons for each site. If the occurrence of few outliers (i.e. high percentages in Fig. 5) is considered as a quality criterion, SCH

shows high-quality data throughout the year. At LAH, high-quality data are possible throughout the year but with highest probability in summer. In addition, LAH shows also the largest variability of percentage of filtered data from all sites, partly because only few winter measurements were attempted at STO and MCO. At STO, only a few outliers were detected in summer data sets, as opposed to winter data (as can be expected from the contact resistances, Figs. 4c and 4d), resulting in a high

filtering rate at step 1 (up to > 50 %), while step 2 excludes all winter measurements (DJF) at STO.

After data inversion, step 3 is exclusion of a data set if its inversion error is greater than a user-defined site-specific threshold (see the threshold values presented in both Table 1 and Fig. S3 in the Supplement). Note, that a low inversion error does not automatically means a good resolution and reliability of the resulting tomogram, as low inversion errors can also be obtained for high filter ratios (Rosset et al., 2013). However, such cases would already be excluded by step 2, which is therefore essential

to keep a high tomogram resolution.

## 3.3  Inversion

We compute the distribution of the specific electrical resistivity for a finite element grid from filtered apparent resistivity data sets. We inverted our data using different inversion schemes and compared the obtained results (similarly to Hellman et al., 2016) and could not detect any significant differences between them that would be relevant for interpretation in a permafrost

monitoring context. In addition, previous studies have also revealed that consistent temporal resistivity changes can be resolved using independent inversions of the data (e.g. Caterina et al., 2017; Hilbich et al., 2011; Oldenborger and LeBlanc, 2018; and Supper et al., 2014), instead of using a time-lapse algorithm (designed to reduce systematic errors for time series with short time gaps, e.g. hours or days). In this paper, in accordance with previous ERT studies at our field sites, the inversion itself is conducted iteratively using the software RES2DINV (Loke, 2018) in batch mode. We use the robust inversion scheme (L1-

norm) with the so-called extended model to minimize the edge effects.

More advanced time-lapse inversion schemes (as proposed e.g. by Karaoulis et al., 2013; Lesparre et al., 2017; Loke et al., 2014) can reduce the occurrence of inversion artefacts which may originate from the under-determined parts of the inversion problems of each individual tomogram. Difference plots of individual tomograms without explicit time-constraints may therefore contain artefacts, especially in regions with low sensitivity. In the following, small-scale anomalies in the resulting

tomograms are therefore excluded from interpretation.

Due to the large number of datasets, as well as the spatio-temporally varying surface and subsurface characteristics at the six sites affecting the RMS, the choice of a consistent and comparable stopping criterion for the inversion is not obvious. For simplicity and based on experience, we here use the third iteration of the inversion, which provides a compromise between an efficient reduction in data misfit and avoiding over-fitting potential measurement noise (e.g. Hilbich et al, 2011).

After the pre-processing and inversion, the tomograms with sufficient quality are selected for further analyses based on a user-defined site-specific threshold for the inversion error (cf. filter step 3). Inversion error histograms (cf. Fig. S3 in the Supplement) illustrate the distribution of the inversion errors for all inverted data sets per site and the site-specific thresholds. As a standardised and non-empirical histogram analysis can indeed not be conducted with such a low number of bins

(statistically, the number of measurements is low cf. Fig. S3), manually defined thresholds based on expert knowledge were used similarly to other studies (cf. Supper et al., 2014).

## 4    Results

### 4.1    Tomograms

The resulting set of accepted ERT tomograms amounts to a total of 901 (out of 1013) for the network. One representative tomogram per site for end-of-summer conditions is given in Fig. 6. Mean resistivity values obtained for all sites span over 3 orders of magnitude as shown in Fig. 3, and highlight the considerable variability of alpine permafrost conditions ranging from fine-grained debris and weathered bedrock at SCH (~ 1 kΩm) to massive ice in a coarse-blocky rock glacier MCO (~ 1000 kΩm). The four other sites exhibit relatively similar resistivity values around 10 kΩm (yellow to light blue colours in Fig. 6).

Note however, that depending on the prevailing lithology, porosity, and subsurface temperature, similar resistivity values can represent very different subsurface characteristics. For example, values around 20 kΩm are interpreted as the unfrozen coarse-blocky layer at MCO and DRE, as ice-poor bedrock permafrost at CER, and as porous ice-rich permafrost at LAH. Except for MCO and CER, all ERTM profiles show significant lateral variability (in the logarithmic scale, cf. Fig. 6).

The thaw depth derived from the respective borehole temperatures is used as validation of the inversion results and is shown

as horizontal lines crossing each borehole in Fig. 6. At CER, LAH and MCO the thaw depth is well represented by vertical resistivity contrasts, while the thaw depth at the ice-poor bedrock sites SCH and STO (plateau) does not necessarily cause significant resistivity contrasts. Here, resistivity variations may rather be influenced by variable ground properties (rock type, degree of fracturation) (e.g. SCH-5000 and STO-6000 boreholes), which can prevent clear relationships between absolute resistivity values and thermal state (frozen/unfrozen).

As the borehole at DRE showed positive summer temperatures during most years of the observation period, no thaw depth is indicated in Figs. 2, 6 and 7. At the time of the ERT measurement shown in Fig. 6, the minimum temperature observed in the borehole was + 1.6 °C (PERMOS, 2016). The resistive anomaly within this talus slope can therefore not (or not entirely) be interpreted as ice-rich permafrost, but rather represents the very porous talus with air-filled voids. The similarity of the resistivity patterns of the talus slopes DRE and LAH is however striking, with ice-rich permafrost confirmed by three boreholes

within the resistive anomaly in LAH (and LAV, not shown). Given the potentially complex pattern of the air circulation process (cf. Wicky and Hauck, 2017) and the undercooled nature of talus slopes (Morard et al., 2010), we cannot fully exclude that parts of the resistive anomaly in DRE may still represent frozen conditions (cf. also similar geophysical studies on low-elevation scree slopes, Gude et al., 2003; Hauck and Kneisel, 2008b). However, with higher probability the zone above the borehole represents a porous blocky layer with smaller block size or more fine material compared to the maximum resistivity

values around the borehole. The DRE tomogram therefore illustrates the potential ambiguity of ERT interpretation in the absence of ground truth or further complementary geophysical data (e.g. seismic data, cf. Kneisel et al., 2008; Maurer and Hauck, 2007).

## 4.2 Long-term resistivity changes

Within our network, ERTM series reach up to 19 years in 2017, which is rare if not unique in the permafrost literature (Supper et al., 2014). Although still short in a climate impact context, they allow the analysis of up to almost two decades of resistivity changes in the context of permafrost thaw and ground ice melt. Figure 7 presents the percentage change of resistivity logarithm

(i.e. $[log10(\rho_2) - log10(\rho_1)]/log10(\rho_1) * 100$) between the first and last year of ERT monitoring for comparable end-of-summer dates for each site. End-of-summer period is considered to be between mid-august to mid-September (except at DRE, where mid-October is the comparison period with the highest number of available data sets). All sites show a significant resistivity decrease within the permafrost layer (below the active layer), except at MCO where a clear trend is less obvious. This is consistent with borehole data, which show increasing permafrost temperatures and active layer deepening at all sites

for the same specific time spans (PERMOS, 2016).

Amongst all sites, the strongest and spatially most consistent resistivity decrease occurs at SCH and LAH, which are also characterized by high permafrost temperatures (see $MAGT_{10m}$ in Fig. 2). From Fig. 7, it can be seen that at LAH and SCH the decreasing resistivity zone reaches down to the bottom of the tomogram (to more than 20 m and 10 m depth, respectively). At LAH, the central part of the resistive body (i.e. between 7 and 15 m depth and from 80 to 140 m horizontally in Fig. 6) exhibits

the smallest temporal resistivity changes, while resistivity decreases above, around and below the resistive body (see Fig. 7). At STO, the resistivity decreased between 4 and 10 m depth on the plateau (left-hand side of the tomogram). Both boreholes situated on the plateau indicate a permafrost degradation (cf. ALT increase in Fig. 7 by 0.9 and 1.7 m respectively), but with a higher rate at the warmer and southern borehole. This is consistent with the thicker negative anomaly at STO-6100 seen in Fig. 7. The strong negative anomaly in the southern slope may also indicate permafrost degradation, but could also be related

to increased water saturation at the time of the measurement.

At MCO, the increase of 0.8 m ALT over the 11-year period is the smallest increase rate of the whole network, which is difficult to resolve on the scale of the ERT resolution (5 m electrode spacing). A comparatively small ALT increase is expected considering the high ice content, and it is also consistent with the results from transient model studies (Marmy et al., 2016; Scherler et al., 2013) predicting a much faster permafrost degradation at SCH compared to MCO during the 21[st] century.

At CER, a significant short-term (during the 4-year observation period) resistivity decrease can be observed along the entire profile at ~ 3-7 m depth, with the strongest decrease in the area between the two boreholes. However, no permafrost degradation trend can be considered over such a short time period, but measurements show a peak in 2015 due to the very warm summer conditions (see Fig. S4 in the Supplement).

At DRE, a significant resistivity decrease can be observed in the middle of the profile (around 50-70 m) at around 5-15 m

depth between 2007 and 2015, which corresponds to the zone interpreted as a blocky layer, which is affected by the air circulation process (cf. Fig. 6). As the borehole indicates unfrozen conditions for both measurement dates (minimum temperature 0.7 °C in 2007 and 1.6 °C in 2015), the positive anomaly can probably not be related to thawing processes, but is

rather expected to represent differences in temperature and humidity related to temporal variability of the complex ventilation process.

## 4.3    Temperature and resistivity relationship for different sites

For the sites, which are measured throughout the year (SCH, LAH and DRE), the seasonal sinusoidal variation of resistivity values demonstrates the general relationship between resistivity and temperature data for the near-surface (Fig. 3, higher resistivity values in winter and lower values in summer). Figure 8 shows selected resistivity-temperature scatter plots for three sites (SCH, LAH and STO), both at shallow depth in the active layer (~ 1-m depth, left panel) and at greater depth within the permafrost (~ 10-m depth, right panel). For the shallow depths, a seasonal signal with regular freezing and thawing is prevailing, which is visible in Figs. 8a and 8c as a well-defined hysteresis loop at SCH and LAH. At STO, non-existing winter data prevent the observation of a similar signal in Fig. 8e. Above the freezing point, the resistivity $\rho$ increases quasi-linearly with decreasing temperature T (see dashed line with a slope of 0.025 K$^{-1}$ in Figs. 8a, 8c) following the equation

$$\rho = \rho_0/(1 + \alpha(T-T_0)) \tag{1}$$

with $\rho_0$ being a reference resistivity at a reference temperature $T_0$ and with $\alpha$ being a temperature coefficient for resistivity given as 0.025 K$^{-1}$ for most electrolytes (Keller and Frischknecht, 1966). The observed hysteresis is caused by the differential behaviour of resistivity during freezing and thawing. During thawing, the resistivity drops abruptly as soon as liquid water becomes available, while the temperature remains at the melting point (zero-curtain effect) during the phase change. At the onset of freezing, liquid water remains present in large quantities enabling electrolytic conduction within the pore space. In addition, ions leaving the frozen phase enhance the pore water conductivity in the liquid phase and decrease the freezing point. Only after sustained freezing, the resistivity starts to increase gradually as temperature further decreases.

Within the permafrost (Fig. 8, right column), seasonal cycles have much less influence and an inter-annual signal with a clear thawing trend with decreasing resistivity values is predominant. At 10 m depth at SCH, the temperature seems to reach a maximum constant value of ~ -0.1°C, which we consider the melting point (see Fig. 8b). The same signal is even more obvious at LAH in Fig. 8d. When temperatures reach the melting point, temperature stays constant, while the ice loss/gain can be followed by decreasing/increasing resistivity data, respectively. Figure 8d shows an example of the zero-curtain at LAH, where the resistivity decreases in autumn 2015, and increases again in spring 2016, without temperature variation. At 11.1 m depth, the seasonal changes of temperature are indeed almost reversed, compared to ground surface temperature, because of the time needed by the cold or warm to propagate into depth. Consequently, resistivity still decreases at the onset of winter.

Figure 8e presents a comparison of resistivity-temperature pairs at both boreholes at STO for two depths within the permafrost (5 and 9 m). Temperatures recorded in the southernmost borehole (STO-6100) are generally higher and closer to the melting point. In both boreholes, at 5-m depth, the variability of temperature and resistivity is higher than at 9-m depth, as it is more influenced by surface variations.  In borehole STO-6000, the recorded temperatures (~ -2.5 °C) at 9-m depth tend to increase with time with comparably small resistivity decrease, while in the southernmost warmer borehole STO-6100 (~ -1°C), the

temperature increase occurs with a significant resistivity decrease, proving the increase of water-to-ice content ratio. At 5-m depth, both boreholes exhibit a resistivity decrease and temperature increase with a much higher variability in resistivity, i.e. a higher variability in liquid water-to-ice content ratio. Due to the large range of values, a logarithmic scale is used in Fig. 8f, although we find a quasi-linear relationship between temperature and resistivity from ~ 0 °C down to our lowest temperature

record (-3 °C). Within this temperature range, surface conduction is enhanced due to the increase in interface area,  whereas in the case of ionic conduction the measured resistivity is influenced by the decreasing amount of liquid water content, leading to decreasing ion mobility and increasing salinity in the liquid phase (Oldenborger and LeBlanc, 2018). In the case of our network sites, the increase of surface conduction is compensated by a rather large decrease in saturation, which explains the resistivity increase. However, the theoretical exponential temperature dependence of the electric current (e.g. Robertson and

MacDonald, 1962; Coperey et al., 2019) is not detectable in our field data, probably because of the narrow temperature range. Similarly, also Krautblatter et al. (2010) observed temperature-resistivity quasi-linearity by laboratory experiments on bedrock samples for sub-zero temperatures between -3 and 0 °C.

## 4.4     Assessment of long-term permafrost degradation

The best example of permafrost degradation detected by ERTM is seen in the almost 20-year time series at SCH. Figure 9

shows the inter-annual evolution of the long-term end-of-summer resistivity changes at SCH,  highlighted by the resistivity logarithm change with respect to the median of all considered SCH data sets (shown for all other stations in the Supplement, Figs. S4 to S8). The resistivity evolution at SCH clearly shows the effect of the heat wave in 2003 (strong resistivity decrease, cf. Hilbich et al., 2008b) and the slow return (until 2008) to the initial state (except for the zone at 2 to 6-m depth and 5 to 15-m horizontal distance, which did not recover its original resistivity value). From 2010 onwards, the ground below 5 m depth

down to the bottom of the tomogram at 15-m depth exhibits even smaller resistivity values than during the record heat wave of 2003. The temporal evolution of both resistivity and temperature at 7-m depth at the SCH-5198 borehole location (see Fig. S9 in the Supplement) shows this evolution as well, but with continuous temporal details. Since 2009, and coinciding with temperatures reaching the melting point for the second time, the resistivity remained low (< 1400 Ωm at 7-m depth) despite substantial ALT variations (maximum 9.5 m in 2015 and 2017; minimum 4.5 m in 2014, see PERMOS, 2016).

In contrast to the effect of the singular heat wave of 2003 (affecting relatively shallow depths), the sustained resistivity decrease since 2009 affects greater depths and extends over the entire length of the ERTM profile. It represents the cumulative resistivity decrease caused by and in agreement with the change in the ground temperature regime since 2009 (warmer summers and winters). The observed ongoing resistivity decrease therefore confirms the sustained increase of liquid water content within the permafrost layer, which is considered a direct consequence of ground ice degradation.

To facilitate inter-site comparison, we plotted in Fig. 10 the resistivity values for each site determined from the spatial average of a manually selected representative zone within the permafrost. Remarkably, all six sites follow a common general trend of long-term resistivity decrease, which is modified by year-to-year variations (e.g. due to meteorological year-to-year variability, irregular dates of the measurement, which may not correspond to the annual maximum thaw depths, or inversion artefacts, see

further discussion below). Note the distinct reductions in resistivity that occur simultaneously between ~ 2009 and 2013 at all sites except CER, where ERTM monitoring only started in 2013.

## 5 Discussion

### 5.1 Reliability and limitations of ERTM

The reliability of ERT results is a constant concern, as inversion results can be ambiguous, especially where model sensitivity is low. Inverting monitoring datasets without an explicit time-constraint in the inversion can produce errors due to noisy data (e.g., Karaoulis et al., 2013; Lesparre et al., 2017; Loke et al., 2014). The objectives of this study include the development and validation of robust inversion approaches that can detect the large-scale climate change signal (permafrost thawing) in the ERT data for very different field sites and conditions, rather than an optimal inversion algorithm for an individual case study. The

consistency of the long-term series (e.g. Figs. 3 and 10) proves the high reliability of most ERT data sets presented in this study, even if single data sets may have a relatively high uncertainty (in case of high electrode contact resistance, high inversion error or random errors). However, regarding the interpretation of temporal resistivity changes, uncertainties may arise from the effects of temporally varying contact resistances and the number of quadrupoles used for inversion, but these are considered small compared to the amplitude of seasonal and long-term resistivity changes. The reliability of single ERT tomograms can

be assessed by either analysing the resolution matrix (Friedel, 2003), the sensitivity uncertainty (e.g. Hermans et al., 2015), or the DOI index (depth of investigation, defined by Oldenburg and Li, 1999), all of which have been applied to data sets obtained on permafrost (e.g. Hauck et al., 2003; Hilbich et al., 2009). Although uncertainties remain, further comparison of resistivity values with borehole data, complementary geophysical data such as refraction seismics (e.g. Maurer and Hauck, 2007; Hilbich, 2010), and hydro-thermal model simulations (e.g. Pellet et al., 2016; Scherler et al., 2010) show a generally good match

regarding the depth of the thaw layer, the layer structure, and its spatio-temporal variability. Moreover, both apparent resistivities and inverted resistivities (Figs. 10a and 10b) show similar decreasing trends, supporting the assumption that existing inversion artefacts are smaller than the long-term measured variations.

Due to the technical difficulties of maintaining several high-altitude monitoring stations and obtaining high-quality data at the same time, our data sets presented in Fig. 3 have major data gaps and most sites are measured only a few times per year.

Although, data gaps in the resistivity time series may prohibit the reliable identification of extreme values related to specific thawing/freezing processes, Hilbich et al. (2008b, 2011) proved the applicability of ERTM in investigating permafrost evolution on different timescales and validated the approach of long-term monitoring with only annual end-of-summer data set (at SCH, before the automatisation of the site). Within the present network, we confirmed the feasibility of using end-of-summer measurements in a long-term evolution context also for morphologically different sites than SCH, and consider our

data sufficient to justify our approach. With respect to resistivity, changes based on the temporal median of resistivity values of the end-of-summer dates (cf. Figs. 9 and 10) permits comparison to sites with only single annual measurements. However, a potential bias might be introduced by variations in the dates used for the end-of-summer measurements alone, or in

combination with variations in the timing of the maximum thaw depth (Hilbich et al., 2011). Having continuously operating ERTM systems, as deployed at SCH, should therefore be the goal for future extensions and new installations of ERTM in permafrost areas.

As mentioned above, a further source of uncertainty may arise from seasonally changing electrode contact resistances and the correspondingly changing number of quadrupoles for the inversions of one ERTM profile. Because the filtering process is applied independently to each data set, the filtered data sets may not consist of the same quadrupoles for each time instance. A time-lapse approach inversion computing temporal differences would therefore be challenging, especially considering the low amount of readings of a Wenner configuration and the low time continuity of our data sets (gaps in time), which may lead to the occurrence of artefacts. However, for the long-term perspective according to the use of end-of-summer measurements as discussed above, this uncertainty is small as differences in quadrupole numbers are small (below 1 % for SCH, LAH and CER and between 5 and 10 % for the other sites for the dates considered in Fig. 7). Moreover, our filtering procedure excludes data sets with more than 20 % filtered quadrupoles (i.e. most winter measurements at STO and LAH, cf. Fig. 5), which avoids extreme cases of comparing inversion results obtained from a different number of readings. However, for detailed process studies on shorter time-scales or at high temporal resolution, such as presented in Hilbich et al. (2011) for SCH, the same number of quadrupoles should be used for inversion to avoid inconsistencies and the occurrence of artefacts in the computed time-lapse differences. In such cases where data sets are derived from continuous monitoring at high temporal resolution, specific time-lapse inversion schemes may likely improve the tomograms significantly (as demonstrated by e.g. Doetsch et al., 2015). In our study, we excluded the interpretation of small-scale anomalies to reduce the uncertainty for the individual inversions.

## 5.2   Temperature-resistivity relations

Hysteresis loops of both directions can be observed within our data set by plotting resistivity against temperature for each borehole location (cf. Figs. 8a, 8c). At SCH, visible hysteresis cycles are similar for both boreholes (13 m distant) close to the surface (0 - 1 m depth, not shown). However, at 1.2 and 1.6 m depth, only borehole SCH-5198 displays a similar (but less pronounced) hysteresis cycle, whereas borehole SCH-5000 exhibits a zero-curtain during freezing that results in a hysteresis cycle in the other direction (see Fig. S10 in the Supplement). In sediments, Tomaškovičová (2018) recorded a zero-curtain effect only during the freezing period, similarly to Fig. 8c but in contrast to Fig. 8a, while a thawing zero-curtain effect similar to Fig. 8a is reported by Hilbich et al. (2011), Pellet and Hauck (2017) and Supper et al. (2014). Different local ground effects contribute to the freezing/thawing hysteresis, e.g. pore connectivity, metastable nucleation, capillarity, and pore blocking (Tian et al., 2014), as well as the mineralogy - the carbonate content being an important factor according to Krautblatter et al. (2010) - are assumed to be responsible for these differences.

On larger temporal scales, the comparison of borehole temperatures and resistivities shows that the last two extreme heat waves in the Swiss Alps - in summers 2003 and 2015 (Scherrer et al., 2016), had a direct effect on ground temperatures and a more sustained effect on resistivities. Heat waves accelerate the decrease of the ice content in the upper part of the permafrost, and

the lost ice is not recovered (air temperatures too high to enable ice aggradation). Therefore, a subsequent warming has a higher effect in the years following a heat wave, as less energy is needed to thaw the decreased amount of ice, leaving more energy for warming of deeper layers (see also Scherler et al. (2013) for a summary of dominant influencing factors for active layer deepening). Figure 9 highlights this effect, which is well documented by the resistivity change below 5 m depth at SCH since 2015.

## 5.3    Overall interpretation of the ERTM network time series

In their review paper on ERTM in the context of permafrost, Supper et al. (2014) compared the acquisition characteristics mentioned in the very few available ERTM studies at that time, and described the SCH installation (Hilbich et al., 2011) as "exceptional". This was because of the favourable conditions in terms of low contact resistances (due to lithology), permafrost temperatures close to the freezing point and low resistivity values on the one hand, and the unusually low ice content in comparison to other ERTM sites on the other. The results of our study confirm this observation in a broader context, with SCH having the lowest resistivities (Fig. 3), the lowest contact resistance values (Fig. 4), and the lowest inversion errors of the entire network. However, the results of the network demonstrate that long-term ERT monitoring is also possible for extremely resistive permafrost landforms such as MCO rock glacier (cf. Fig. 10a) with much higher contact resistances, but not significantly higher inversion misfits, at least for the summer period. Figure 10c also indicates that analysing several sites simultaneously with a similar methodology helps to increase the representativeness of observed changes in ground ice occurrences. With the ERTM network presented in this study, the SCH data can now be set in relation to ERTM results from other permafrost landforms such as rock glaciers and talus slopes.

The parallel evolution of mean resistivity within the permafrost layer in all parts of Switzerland (and the Italian Alps) confirms the high representativeness of ERTM data for the analysis of ongoing permafrost degradation. Importantly, this spatially averaged resistivity trend represents a much larger footprint of permafrost conditions than would otherwise have been determined according to the point information of borehole temperature profiles (cf. the standard PERMOS temperature time series plots shown in the Supplement, Fig. S11). Furthermore, ERTM results are directly relatable to ground ice melt without the masking effect found in temperature series due to latent heat release close to the melting point. The observed parallel resistivity evolution at all sites therefore clearly illustrates the ongoing permafrost degradation with a remarkable shift to another resistivity regime after 2009, in agreement with temperature evolution.

In addition to the top-down permafrost degradation observed at all sites, Fig. 7 shows a resistivity decrease at LAH below the ice-rich body.This may be explained by air circulation in the talus slope, particularly below the ice-rich resistive body (Delaloye and Lambiel, 2005; Wicky and Hauck, 2017), which might lead to bottom-up permafrost degradation. Thinning of the permafrost body from below is indeed confirmed by increasing temperatures around 20 m depth (the sensor at 21.5 m depth in borehole LAP-1108 measured constantly negative temperatures until 2012, and is recording positive temperatures since 2015 (cf. PERMOS, 2016)). The observed resistivity decrease between 15 and 20 m depth, although close to the limit of the

sensitivity of the tomogram, may therefore result from an increase in the ratio between liquid water content and ice content or increasing ion mobility in the liquid pore water at negative temperatures (Oldenborger and LeBlanc, 2018).

## 5.4    Permafrost monitoring framework

The ERTM network is part of a broader permafrost monitoring framework, which includes other variables like air and ground temperature (e.g., Isaksen et al., 2011; PERMOS, 2019), as well as soil moisture (Pellet and Hauck, 2017; Wang et al., 2016), P-wave velocity (Hilbich, 2010; Krautblatter and Draebing, 2014), and permafrost creep velocity (Delaloye et al., 2010; Kellerer-Pirklbauer and Kaufmann, 2012; PERMOS, 2019). Figure 11 shows the simultaneous temporal evolution of electrical conductivity (the inverse of electrical resistivity) together with other monitoring variables from different sites in Switzerland, chosen for their long time series. We present here the conductivity instead of the resistivity for visual consistency with the trend of all other parameters. Interestingly, the increasing trends in rock glacier velocity and the ground temperature show a break in 2016, while the conductivity trend continues to increase.

Similar trends are also observed at almost all other PERMOS field sites and they represent general tendencies for the entire Swiss Alps (PERMOS, 2019). The observed decrease in ground temperature and creep velocity in 2017 is due to the late onset of snow during the 2016/17 winter, which enabled a more efficient cooling of the ground. This effect is not seen in the resistivity data at SCH, because only end-of-summer measurements are considered. Deep ground cooling also occurred during that winter at SCH (shown in Fig. S9 in the Supplement), leading to a seasonal increase in the ice to water content ratio. However, this increase was countered by the especially warm spring and summer until the end-of-summer measurement in 2017. The temperature measured at 9 m depth in summer 2017 was indeed the second highest after 2015. The decrease in temperature in 2016/17 was more pronounced at sites with thinner snow cover (and late snow onset) and colder permafrost temperatures like at MCO (see Fig. S11 in the Supplement), and at STO and LAH where it caused a small increase in the resistivity in 2017 (Fig. 10).

The fact that all measured variables in Fig. 11 (and similar data for other stations in the Alps) exhibit the same long-term evolution of permafrost, i.e. degradation due to climatic conditions, is very likely due to the current disequilibrium between atmospheric and ground thermal conditions. The resulting permafrost degrading trend is so dominant, that it outweighs any impacts of morphological, topographical, climatological, or geological differences between the sites.

## 6    Conclusions and Outlook

This study presents the analysis of a network of six electrical resistivity tomography monitoring (ERTM) sites in the central Alps, where at least one measurement per year at the end-of-summer was conducted for more than one decade (except CER, which has only five years of ERTM data). The network data is comprised of 1013 data sets in total until the end of the year 2017. Systematic data processing includes a) the analysis of contact resistances, and b) the application of automatic processing routines (filtering, data inversion, and time series analysis), which allows for a comparative analysis of the long-term resistivity

evolution in the context of climate induced permafrost degradation. Following previous studies in ERT monitoring on mountain permafrost, we used independent inversions instead of time-lapse regularization of difference-inversions to provide a general overview of the long-term trend in the electrical parameters.

The ERTM network covers representative permafrost landforms, including warm permafrost with high temperatures and low ice content, and cold permafrost with massive ice as well as a high spatio-temporal variability of ground ice occurrences. All permafrost sites show a clear permafrost degradation signal, which is strongest for the longest time series at SCH, regarding different parameters: temperature increase, active layer thickening, and resistivity decrease within the permafrost body (i.e. liquid water to ice content ratio increase). The correlation between temperature increase below the freezing point and resistivity decrease is significant at all sites and is especially pronounced around the permafrost table. Moreover, the observed degradation trend is particularly visible between ~ 2009 and 2013. In spite of the wide range of morphological, climatological, and geological differences between the sites, the observed inter-annual resistivity changes and long-term tendencies are similar for all permafrost sites. The current permafrost degradation (over one to two decades) is evidenced by both borehole temperature time-series and electrical resistivity time-series.

Despite the difficulty to maintain long-term ERTM sites in mountain permafrost terrain, we believe the effort to be very valuable, as long-term time series are of high importance regarding permafrost monitoring and the understanding of the impacts of climate change and meteorological anomalies. ERTM provides the possibility to detect and monitor ground ice loss, even when borehole temperatures stagnate at the melting point (zero curtain effect). ERTM further helps to evaluate the representativeness of borehole temperature information for the entire landform. Furthermore, this paper demonstrates the advantages of combining data from several sites into an ERTM network. Similar to the existing global temperature network of permafrost GTN-P (Biskaborn et al., 2015), the ERTM installations currently initiated around the world could lead to a global ERTM database and used as a complementary assessment of ongoing permafrost change on a global scale.

The application and subsequent validation of a more advanced inversion strategies (e.g. 4D) is beyond the scope of the present paper. In a forthcoming study, such an approach could first be applied at SCH, where resistivity data at high temporal resolution is available (Hilbich et al., 2011). This data availability is, however, the exception at operational permafrost monitoring stations, which currently favours a simple and robust data processing approach that is feasible for operational monitoring networks. Further electrical methods such as (spectral) induced polarization (e.g., Flores Orozco et al., 2018) and spontaneous potential (e.g. Slater, 2007) may help to overcome limitations of the ERT method. However, applications of these methods to permafrost are still rare (e.g. Doetsch et al., 2015; Duvillard et al., 2018; Grimm and Stillman, 2015) and no long-term monitoring data sets exist yet.

## Acknowledgments

We are thankful to the PERMOS office for providing the temperature data and supporting our monitoring work and to all field helpers. We are grateful to the cable car companies Schilthornbahn AG, Télénendaz S.A. and Corvatsch AG for the logistical support to our research activities. We also thank our colleagues from ARPA (Italian Regional Environmental Protection Agency) for supplying the data from the Cervinia monitoring site. We acknowledge Oliver Sass, Andre Revil, Sara Bazin and the editor Peter Morse for their helpful comments and suggestions that improved the manuscript quality.

## Data availability

The borehole temperature data from the PERMOS network are published under PERMOS (2016), https://doi.org/10.13093/permos-2016-01. The ERT data can be obtained on request from the authors (ERT data from Schilhtorn, Stockhorn, Lapires and Murtèl will also be available in the near future on the PERMOS website (http://www.permos.ch).

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

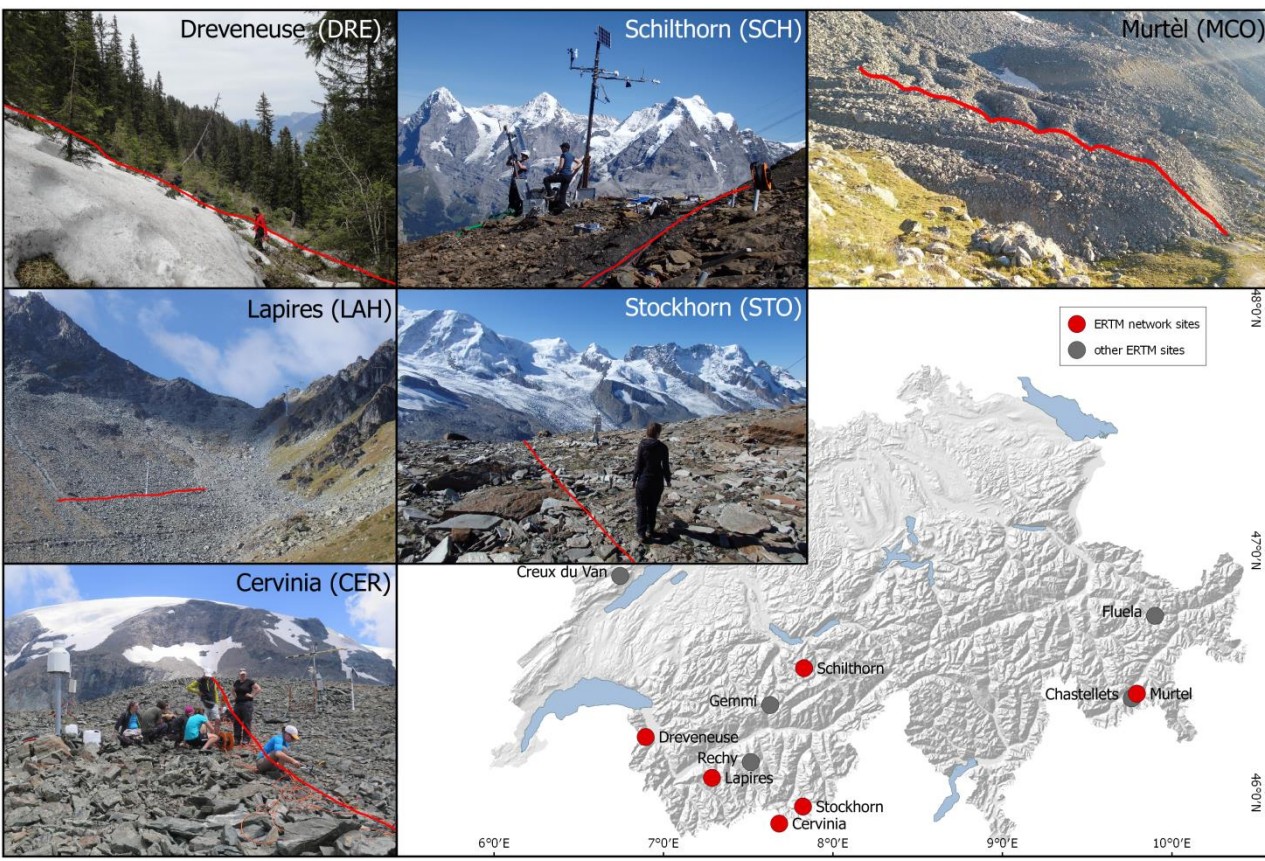

**Figure 1 Map of Switzerland with location of the ERTM network sites presented in this paper (in red) and other monitoring sites (in grey). Photographs show the location of the ERT profiles in red for each site of the network. Note that CER is located in Italy.**

Table 1 Key information of the geophysical monitoring installations.

| Site | Altitude (m asl) | Start year | Number of electrodes | Config. | Acquisition system | Spacing (m) | Profile length (m) | Maximum depth of investigation (m) | Number of data sets (until 2017) | Total number of measured data points | Inversion error threshold (%) |
|---|---|---|---|---|---|---|---|---|---|---|---|
| SCH | 2910 | 1999 | 49 (30) | W | automatic | 2 | 96 | 14 | 857 | 224 868 | 4.5 |
| STO | 3410 | 2005 | 50(48; 55) | W | manual | 2 | 98 | 18 | 36 | 14 656 | 10 |
| LAH | 2500 | 2006 | 43 (35) | W | autom./man. | 4 | 168 | 30 | 65 | 16 810 | 15 |
| MCO | 2670 | 2006 | 48 | W | manual | 5 | 235 | 40 | 23 | 8 280 | 15 |
| DRE | 1600 | 2007 | 48 | W, WS | manual | 3 | 141 | 25 | 38 | 21 689 | 10 |
| CER | 3100 | 2013 | 48 | WS | manual | 2 | 94 | 18 | 7 | 3 703 | 6 |
| CDV[1] | 1240 | 2007 | 48 | WS | manual | 5 | 235 | 40 | 26 | 13 754 | |
| FLU[1] | 2450 | 2009 | 50 (75) | W | manual | 4 | 296 | 30 | 2 | 1 292 | |
| GFU[1] | 2450 | 2013 | 48 | WS | manual | 2 | 94 | 25 | 4 | 1 440 | |
| LAV[1] | 2500 | 2007 | 70 | W | manual | 4 | 276 | 25 | 28 | 17 584 | |
| MF[1] | 2600 | 2009 | 48 | W | manual | 1 | 47 | 8 | 18 | 6 480 | |
| REC[1] | 2700 | 2008 | 71 | W, WS | manual | 4.5 | 315 | 30 | 15 | 11 055 | |
| RC[1] | 2600 | 2009 | 72 | W | manual | 1 | 71 | 8 | 18 | 11 574 | |
| SCV[1] | 2910 | 2006 | 47 | W, WS | automatic | 4 | 184 | 32 | 227 | 78 315 | |

Number of electrodes in an earlier setup is specified in brackets. W: Wenner array; WS: Wenner-Schlumberger array. Data
source: Mewes et al., 2017; Morard, 2011; Pellet et al., 2016; PERMOS, 2016.
[1]: ERTM sites which are not in the focus of the present paper due to (i) absence of borehole information or (ii) interruption of
measurements for safety reasons. CDV: Creux du Van (Delaloye et al., 2003); FLU: Flüela (Kenner et al., 2017); GFU: Gemmi
(Pellet et al., 2016); LAV: Lapires vertical; MF: Murtèlforefield (Schneider et al., 2013); REC: Becs-de-Bosson/Réchy (Mewes
et al., 2017); RC: Chastellets (Schneider et al., 2013); SCV: Schilthorn vertical (Noetzli et al., 2008).

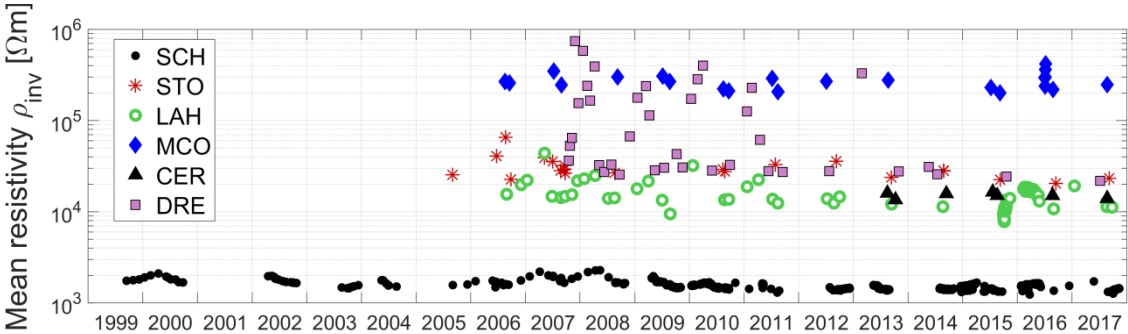

**Figure 2** Temperature key information of the monitoring sites, including mean annual air temperature (MAAT), active layer thickness (ALT) of 2017, mean annual ground temperature at 10 m depth (MAGT10m), and the average resistivity (ρ10m) at around 10 m depth at the borehole location (extracted from all available data sets), a depth where temporal variability and corresponding errors from discontinuous measurements are small. Where available, data are given for two boreholes (1 and 2). No ALT data exist for DRE due to unfrozen conditions in the borehole. MAAT and borehole data from: PERMOS database (PERMOS, 2016), Arenson et al. (2010) and Pogliotti et al. (2015).

**Figure 3** Overview of the available ERTM time series for the six stations of the network and comparison of their spatial mean resistivities.

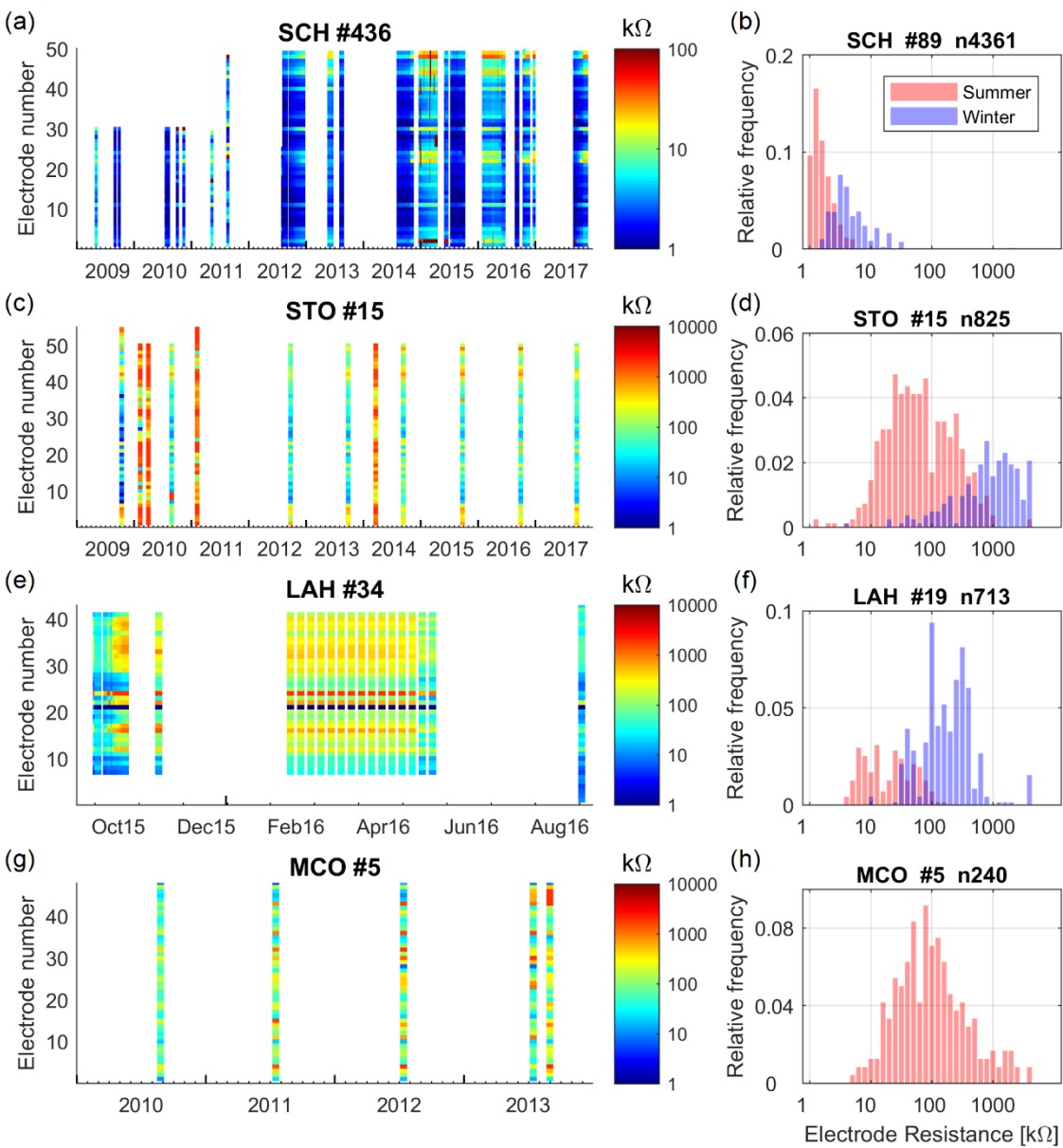

**Figure 4 Temporal evolution of contact resistances and seasonal variations of contact resistance histograms at (a,b) SCH, (c,d) STO, (e,f) LAH and (g,h) MCO. Summer (red) is hereby defined as the three months with thawing processes and an absence of snow cover (JAS), while January to March (JFM) represent the winter conditions (blue). No winter values are available for MCO. Note the different time and colour scales. The symbol "#" denotes the number of data sets, and "n" the total number of electrodes taken into account in the histograms.**

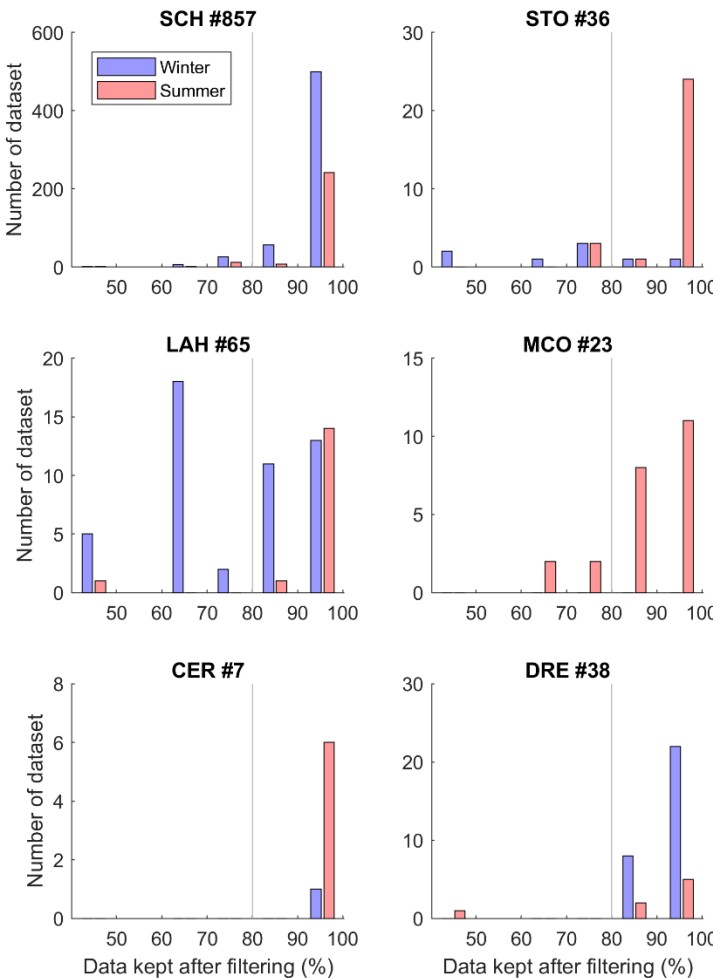

**Figure 5 Seasonal analysis after filter step 1for all sites. Step 2 consists in a cut-off filter of 80% (grey line). Summer (red) is here defined as the three months with thawing processes and an absence of snow cover (JAS), and winter conditions (blue) are represented by the nine other months (October - June). The symbol "#" denotes the number of data sets taken into account for each site.**

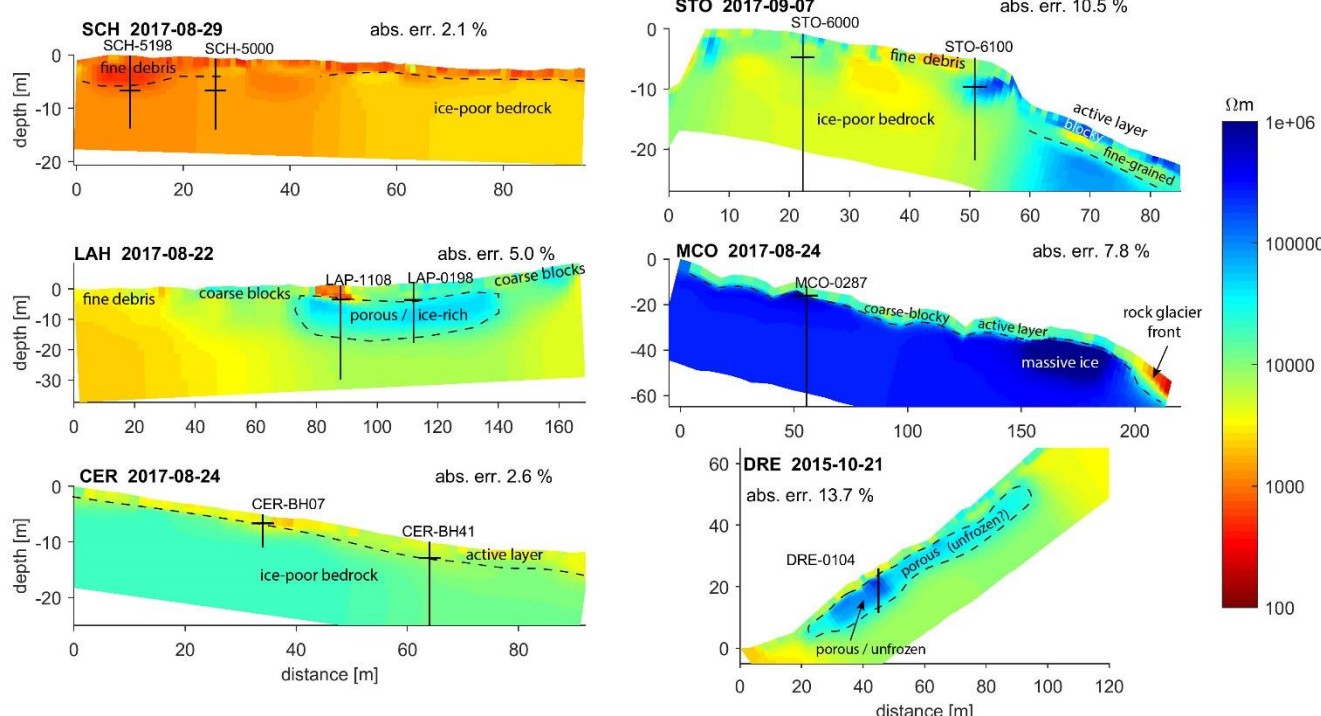

**Figure 6 Representative end-of-summer tomograms for each site with interpretation of major characteristics. Boreholes along the resistivity profiles are represented in black, with horizontal lines indicating the thaw depth at the date of the ERT measurement, except for DRE, where the borehole is completely unfrozen at the time of ERT measurement. Note, a second line for LAH at borehole LAP-1108 has been added to represent the permafrost base limit.**

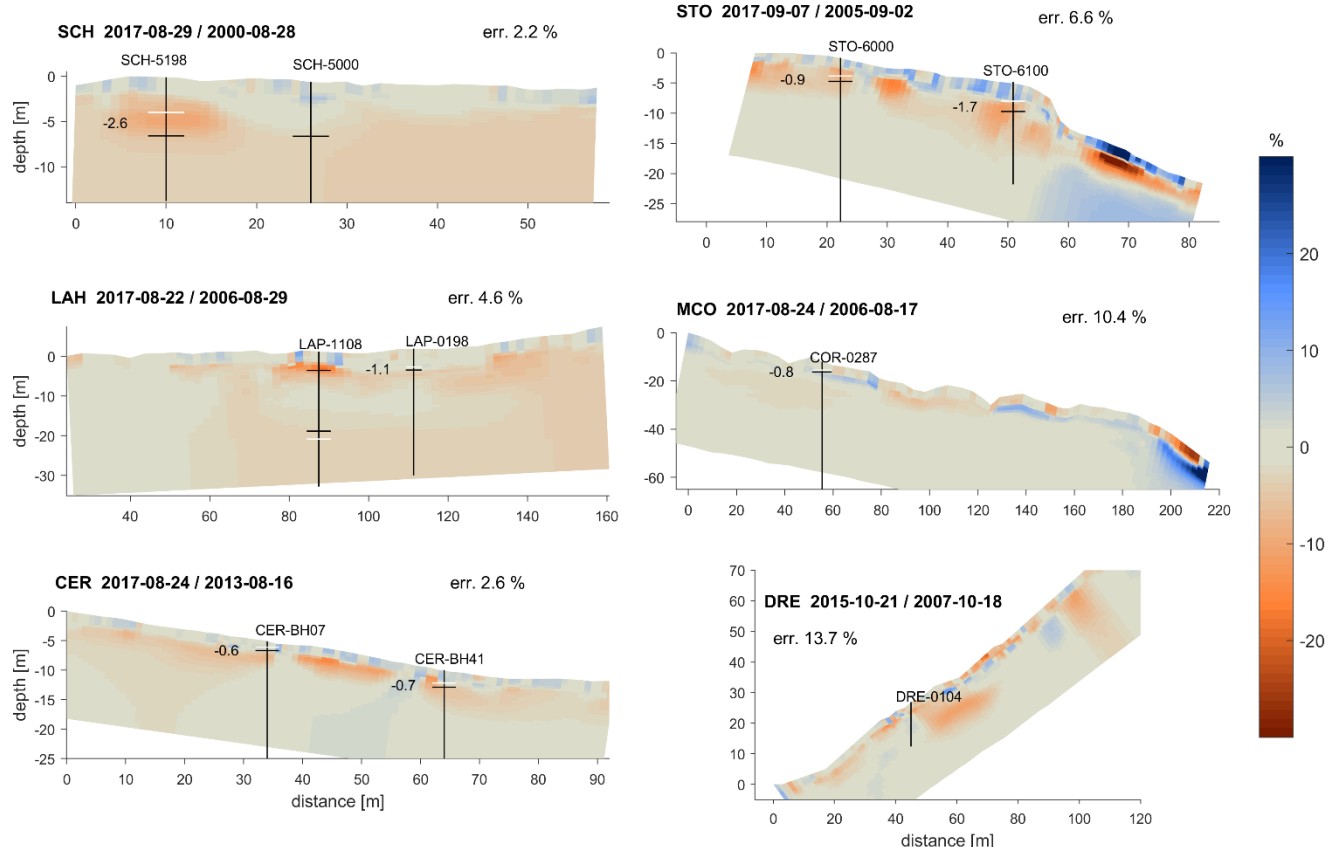

**Figure 7 Percentage change of resistivity logarithm over the full observation period for comparable dates at end of summer for each site. Vertical lines represent borehole positions and horizontal lines the thaw depth (white and black lines correspond to the oldest and most recent date, respectively). Change in ALT is expressed in metres near the boreholes. Missing white lines (for SCH and LAH) correspond to non-existing data.**

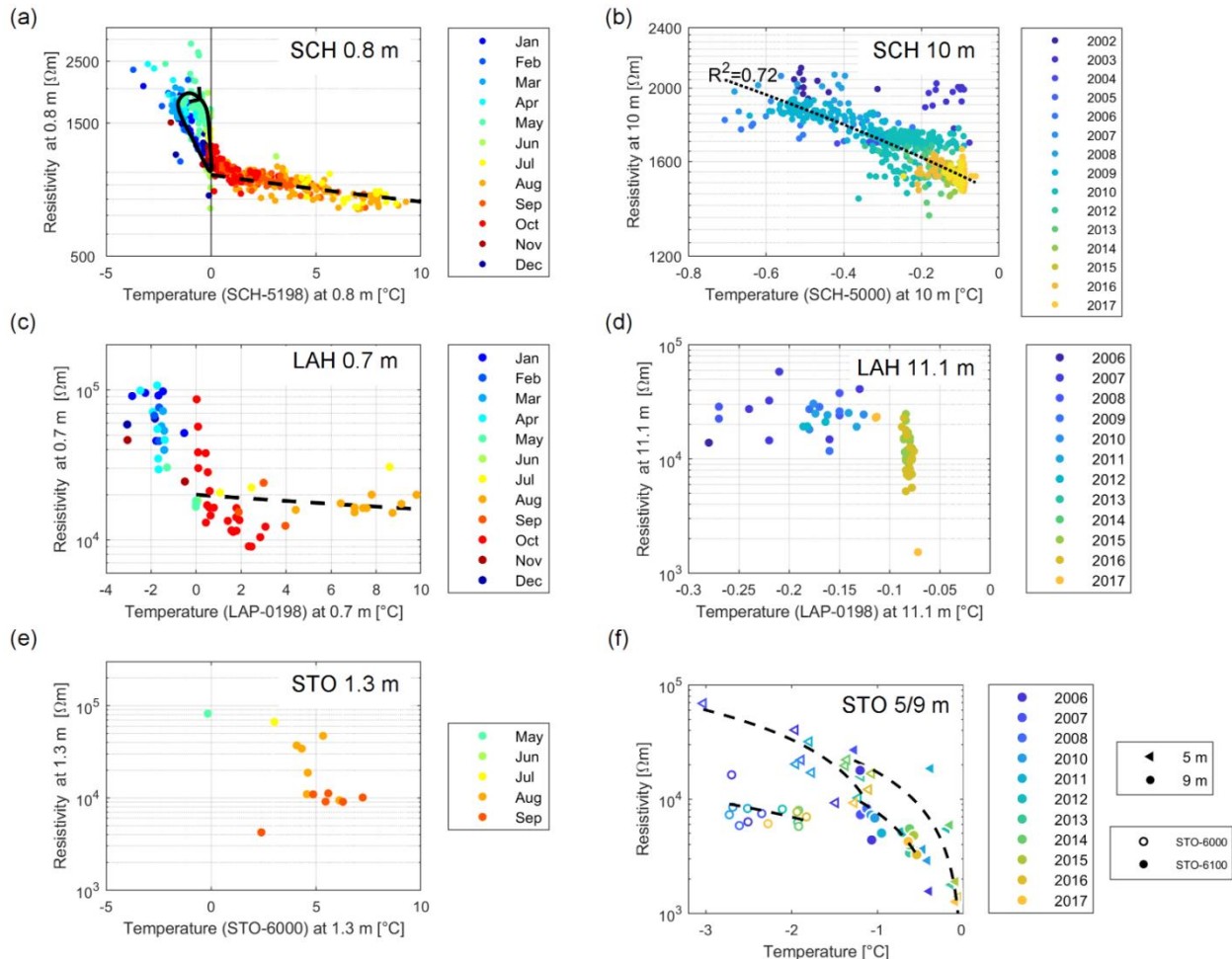

**Figure 8 Example of resistivity-temperature scatter plots at SCH, STO and LAH at shallow depth (left panel, where the colour scale denotes months) and within the permafrost (right panel, where colour scale denotes the years). (a) SCH at 0.8 m depth, where a hysteresis loop is observed (direction indicated by black arrow) at shallow depth. The dashed line shows the linear dependency above the freezing point (0.025 K$^{-1}$). (b) SCH at 10 m depth: The dotted line corresponds to a linear regression. (c) LAH at 0.7 m depth: The dashed line shows the linear dependency above the freezing point (0.025 K$^{-1}$). (d) LAH at 11.1 m depth. (e) STO at 1.3 m depth and (f) STO for both boreholes, STO-6000 (unfilled symbols) and STO-6100 (filled symbols). Symbols denote the depth. Dashed lines fit the data for each depth and each borehole.**

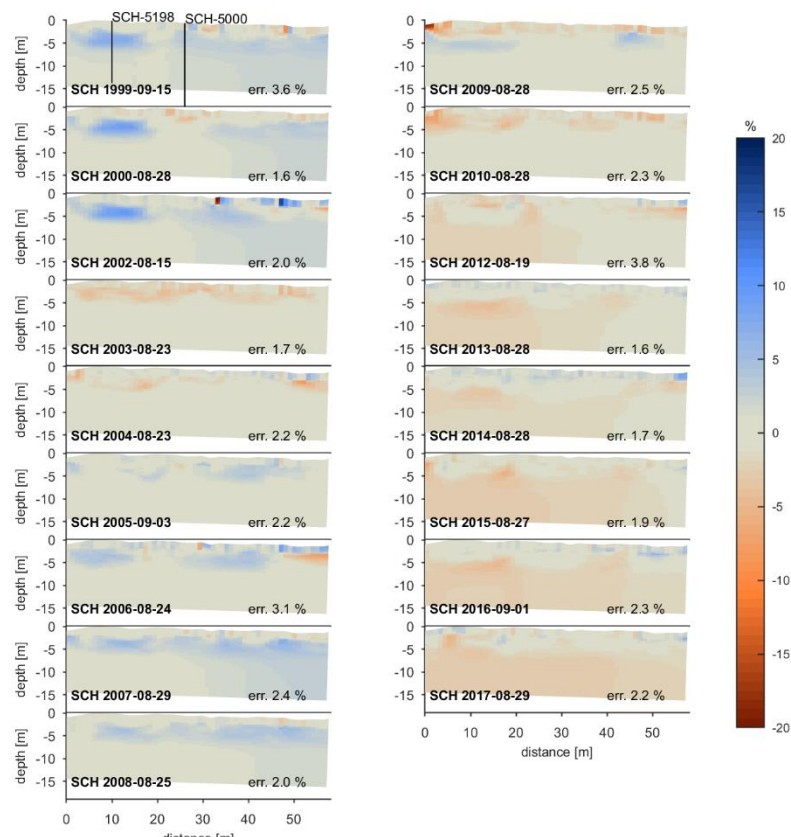

**Figure 9 Resistivity logarithm percentage change tomograms with respect to the median of the end-of-summer time series at SCH. Black lines represent borehole positions.**

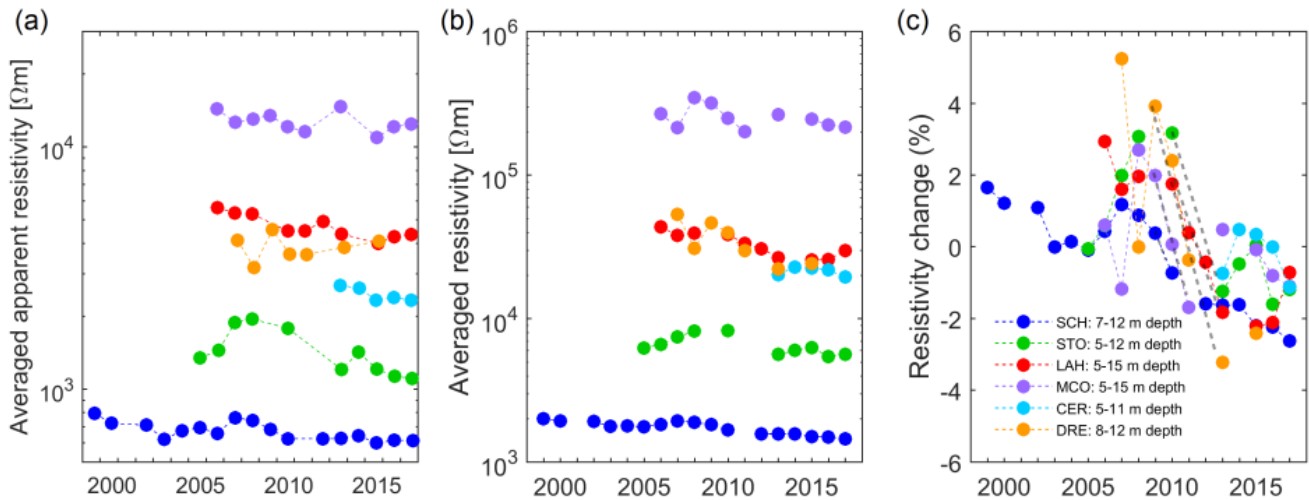

**Figure 10** Time series of the end-of-summer permafrost selected zone of (a) spatially averaged apparent resistivity (b) spatially averaged resistivity logarithm and (c) logarithm resistivity changes relative to the median of the end-of-summer time series. End-of-summer represents the second half of August, with extreme dates ranging between mid-July and mid-September for years without end-of-August dataset. The grey dashed lines highlight the period (~ 2009-2013) of remarkable resistivity decrease at all sites.

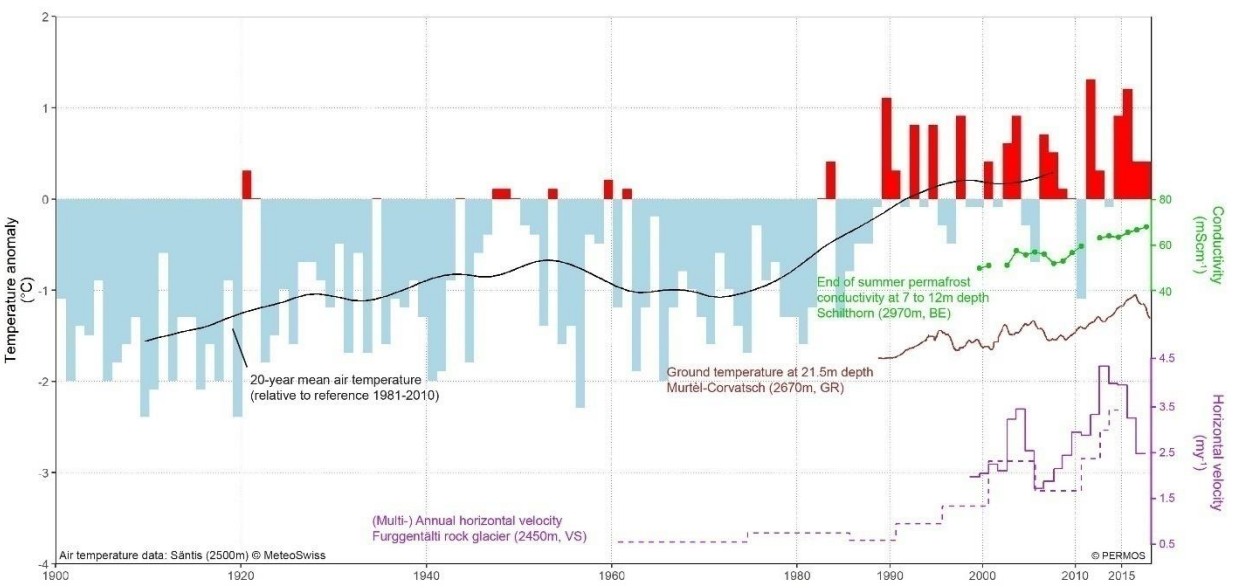

**Figure 11** Evolution of the permafrost temperature (brown line), rock glacier horizontal velocity (purple line) and permafrost electrical conductivity (green dots and line) at selected PERMOS field sites in the context of long-term climate observations. Annual air temperature anomaly (relative to the 1981-2010 norm period) measured at Säntis (2500 m) is shown by blue and red bars, and its smoothed value is shown in black. The dashed purple line represents horizontal rock glacier velocity reconstructed from orthophotos (PERMOS, 2019).