# Peer review of "Mountain permafrost degradation documented through a network of permanent electrical resistivity tomography sites"

_The Cryosphere, 2018_

## Referee Comment (RC1) · Oliver Sass (Referee) · 3 Mar 2019

This contribution reports on permafrost monitoring using repeated 2D electrical resistivity measurements at six sites in the western Alps. The huge dataset is unique and highly valuable which definitely warrants the publication in The Cryosphere. What is more, the paper is structured clearly and flawlessly written. I particularly liked the excellent, comprehensible part on error analysis and filtering. Thus, I recommend publication after some very minor revisions.

General comments:

[Figure]

The authors are somewhat too modest with regard to their formulated aims. p.3: " The purpose is to demonstrate the benefit of joint ERTM data processing and interpretation within a regional network as complementary 2-dimensional information about freezing and thawing processes in mountain permafrost compared to point-scale borehole data." This is rather self-explaining. Nobody will question the benefits. Maybe add additional aims, e.g.: Assess permafrost degradation over a long time span Investigate the correlation between temperature increase and resistivity decrease Explore differences between geologically and climatically different sites

P14 L5-20 should be shifted to the Discussion section

In the Discussion, mainly Swiss authors from the working groups of the authors are cited; you should add some more references to authors from other countries.

Editorial comments:

P9 L20: one blank missing P12 L29: correct: Fig. 8, right panel

———————————————

---

## Referee Comment (RC2) · Andre Revil (Referee) · 7 May 2019

Review of "Long-term monitoring of mountain permafrost degradation using an electrical resistivity tomography network". This manuscript presents an interesting dataset by Mollaret et al. of long-tern electrical resistivity tomography for permafrost under the effect of climate change. The raw dataset itself is very impressive. The manuscript is however weak regarding the type of time-lapse inversion used (no use of time, which is in my opinion a drastic error, see discussion below) to produce the end results as well regarding the poor discussion of the underlying physics of electrical resistivity / temperature changes in this situation. These are the two main weaknesses of the

current manuscript, which should be however published but the current manuscript is half-baked.

1. First I think a better analysis of the underlying physics of electrical conductivity / temperature / water content relationships under freezing conditions is required. See for instance Duvillard P.A., A. Revil, A. Soueid Ahmed, Y. Qi, A. Coperey, and L. Ravanel, Three-dimensional electrical conductivity and induced polarization tomography of a rock glacier, Journal of Geophysical Research, 123, . https://doi.org/10.1029/2018JB015965, 2018. Anotehr paper ion this topic is in press in JGR-SE. There are effects associated with surface conduction for instance.

2. ERTM (i.e. monitoring) is only superior to ERT if the time is properly accounted for in the inversion e.g., through sequential inversion or real .5D+time (or 4D° inversion i.e., including reguilarization over time as it is done over space (see for instance Karaoulis M., A. Revil, D.D. Werkema, B. Minsley, W.F. Woodruff, and A. Kemna, Time-lapse 3D inversion of complex conductivity data using an active time constrained (ATC) approach, Geophysical Journal International, 187, 237–251, doi: 10.1111/j.1365-246X.2011.05156.x, 2011; Karaoulis M., A. Revil, D.D., Werkema, P. Tsourlos, and B.J. Minsley, IP4DI: A software for time-lapse 2D/3D DC-resistivity and induced polarization tomography, Computers & Geosciences, 54, 164-170, 2013.). Such a discussion is required, inverting time lapse dataset without the inclusion of time in the inversion can lead to serious errors due to noisy data and artefacts in the inversion process itself (do inversion at two different iteration at two different times will produced many spurious noamalies). The ATC approach can handle strong change in the data space so the argument "This is because our measurements are characterized by irregular time gaps and by strong spatial and temporal resistivity contrasts (active layer frozen/unfrozen)" is, in my opinion, incorrect.

3. Personally I never accepted that reciprocity is necessarily the best test for error quantification and outliers removal. But I am fine with that. In principle, real time lapse inversion with the ATC technique filters out the data that are not correlated over time.

4. RES2DINV and the L1 option is a very bad choice for inversion. The algorithm tends to produce step-like structrures when the number of iteration in the inversion process goes being 3. These are inversion artefacts. So this is bit sad that such an amazing dataset is inverted in a careless way. In addition, RES2DINV does not take into account the data covariance matrix. A big drawback when dealing with noisy data.

5. Section 4.1. what about induced polarization? Self-potential?

6. The approach for looking at time lapse change in resistivity is very dangerous because the resistivity data have not been inverted with appropriate time-lapse regularization techniques. The relevant literature on the subject is even not cited and it is has been shown in the literature that doing like this brings a lot of artefacts in the computed changes because of inversion artefacts.

7. Section 4.3 is far from our level of knowledge regarding the underlying physics of the relationship between resistivity and temperature, which is non-linear, see Duvoiillard et al. 2018 and Coperey et al. 2019 (both in JGR-SE).

A.Revil May 7th, 2019

---

## Short Comment (SC1) · 29 May 2019

Review of "Long-term monitoring of mountain permafrost degradation using an electrical resistivity tomography network" by Mollaret et al.

This manuscript reports results of 2D ERT monitoring surveys at six sites in the western Alps. The authors draw conclusions on the state and the degradation of permafrost likely related to climate changes, which deserves publication in The Cryosphere.

This paper is very well written and the figures are well presented. The abstract is concise and the introduction gives a good overview and presents the motivation. This

long-term dataset is extremely valuable. The data quality is variable and the data selection is well explained. The data inversion seems well done. The way to illustrate the seasonal effect in Figures 4 and 8 is very nice. Figure 10 is a very clever and innovative way to illustrate the long-term variation.

The amount of work to compute these tomographic images is probably enormous and the results are impressive. The analysis is thorough and the interpretation is very interesting. Thus, I recommend publication and I have very few comments to improve this manuscript.

The only requests/suggestions would be: - Add the reciprocal measurements in the supplement. It is a bit useless to talk about them, without showing figures. - Check the format for referencing several publication in the text. I understand that the rule is chronologically before alphabetically. - Add a more general paragraph on ERTM time series. I feel that some references on resistivity monitoring are missing.

Comments: Page 2 Line 2: Change "may" by "will" Page 4 – line 4: ALT is already explained earlier Page 4 line 18: explain the short and long-wave radiations for those who are not familiar with meteorological stations. Page 6, line 28: I would remove "Here, . . ." Page 9, line 22. Please add the user-defined site-specific thresholds for the inversions in table 1. Page 9: I would rephrase or expand the sentence in line 25. Page 10: I would also rephrase or expand the sentence in lines 9-12.

Figure 2: I would not use blue for ALT1 and ALT2. It is confusing with the 2 other blues (LAGT10m 1 and 2). In the caption, add "Where available, data are given for two boreholes (1 and 2). Figure 10: I really like these 2 figures. The right one is however a bit difficult to understand/follow. The yellow background is not a good choice. Maybe you could use thicker and coloured lines in this area.

Sara Bazin Oslo, 27/05/2019

---

## Author Comment (AC1) · 28 Jun 2019

The comment was uploaded in the form of a supplement:
https://www.the-cryosphere-discuss.net/tc-2018-272/tc-2018-272-AC1-
supplement.pdf

---

## Author Response (AR1)

Dear editor and reviewers,

We thank you sincerely for your constructive and helpful comments of our manuscript. We have modified it accordingly to address your suggested revisions. Please find below a point-by-point response. All modifications are marked in red in the revised manuscript. We hope that our corrections properly address your concerns.

Sincerely,

Coline Mollaret on behalf of all authors

Referee #1 comments appear in blue,
Referee #2 comments in green,
Short comment #2 in red,
Author responses in black,
and original text and the suggested revision appear in grey.
Please find below our detailed answers to all reviewer comments.

Author response to reviewer comments:

**Referee #1: Review from Oliver Sass (March, 3rd 2019)**

This contribution reports on permafrost monitoring using repeated 2D electrical resistivity measurements at six sites in the western Alps. The huge dataset is unique and highly valuable which definitely warrants the publication in The Cryosphere. What is more, the paper is structured clearly and flawlessly written. I particularly liked the excellent, comprehensible part on error analysis and filtering. Thus, I recommend publication after some very minor revisions.

We thank the reviewer for the positive view and comment.

General comments:

The authors are somewhat too modest with regard to their formulated aims. p.3: "The purpose is to demonstrate the benefit of joint ERTM data processing and interpretation within a regional network as complementary 2-dimensional information about freezing and thawing processes in mountain permafrost compared to point-scale borehole data." This is rather self-explaining. Nobody will question the benefits. Maybe add additional aims, e.g.: Assess permafrost degradation over a long time span Investigate the correlation between temperature increase and resistivity decrease Explore differences between geologically and climatically different sites

We thank the reviewer for the constructive suggestions. We followed the recommendation of the reviewer by adding the following sentence at the end of the Introduction section:

New version:

The purpose is to assess permafrost degradation over one to two decades, investigate the correlation between temperature increase and resistivity decrease, and explore differences between geologically and climatically different alpine sites.

P14 L5-20 should be shifted to the Discussion section

We followed the recommendation of the reviewer and the mentioned paragraph is now in the discussion section (at the end of section 5.3).

In the Discussion, mainly Swiss authors from the working groups of the authors are cited; you should add some more references to authors from other countries.

We thank the reviewer for raising this point. As we present a data set from a regional monitoring network, an important part of the discussion is of course focused on validation of our results with other complementary data from the same sites, which causes a regional focus in the references. Some references to non-Swiss groups (e.g. Supper et al (2014), Tomaskovicova (2018), Krautblatter et al. (2010)) were already present. But, as suggested and for a more balanced discussion, we added references to the following non Swiss ERTM studies: Doetsch et al., 2015; Karaoulis et al., 2013; Kellerer-Pirklbauer and Kaufmann, 2012; Lesparre et al., 2017; Loke et al., 2014.

Editorial comments:

P9 L20: one blank missing
P12 L29: correct: Fig. 8, right panel

We corrected it accordingly. We thank the reviewer for the careful reading.

**Referee #2: Review from Andre Revil (May, 7th 2019)**

Review of "Long-term monitoring of mountain permafrost degradation using an electrical resistivity tomography network". This manuscript presents an interesting dataset by Mollaret et al. of long-tern electrical resistivity tomography for permafrost under the effect of climate change. The raw dataset itself is very impressive. The manuscript is however weak regarding the type of time-lapse inversion used (no use of time, which is in my opinion a drastic error, see discussion below) to produce the end results as well regarding the poor discussion of the underlying physics of electrical resistivity /temperature changes in this situation. These are the two main weaknesses of the current manuscript, which should be however published but the current manuscript is half-baked.

We thank the reviewer for his appreciation of the data sets of the network, which are the core of the current paper, and for the detailed comments regarding the underlying physics and inversion, which we will address point-by-point below.

1. First I think a better analysis of the underlying physics of electrical conductivity / temperature / water content relationships under freezing conditions is required. See for instance Duvillard P.A., A. Revil, A. Soueid Ahmed, Y. Qi, A.Coperey, and L. Ravanel, Three-dimensional electrical conductivity and induced polarization tomography of a rock glacier, Journal of Geophysical Research, 123, .https://doi.org/10.1029/2018JB015965, 2018. Anotehr paper ion this topic is in press in JGR-SE. There are effects associated with surface conduction for instance.

We improved the description of the underlying physics of electrical conductivity and added references accordingly in several sections of the manuscript as follows:

Section INTRODUCTION:
Old version (P2, L20-23):

As electric conduction in soils/rocks is mainly controlled by ionic charge transport taking place through water-filled pores, and to a lesser extent by surface conduction mechanisms at mineral-water interfaces, spatial and temporal resistivity variations in the subsurface depend on porosity, pore connectivity, mineralogy, water chemistry and temperature (e.g. Hermans et al., 2014; Ward et al., 2010).

New version:

The electrical resistivity (or its reciprocal, the electrical conductivity) combines the contribution of three conduction mechanisms: particle conduction, surface conduction and pore fluid conduction (e.g. Klein and Carlos Santamarina, 2003). Particle or electronic conduction is mainly related to metallic materials (due to the high number of free electrons). Surface conduction mechanisms take place at the electrical double layer formed at the mineral-ice-water interface (at the electrical double layer) and is controlled by the water content and the electrical properties of the interface (pH, zeta-potential, cation exchange capacity). Finally, the ionic or electrolytic conduction takes place through fluid-filled pores and is controlled by porosity, saturation, pore connectivity, and fluid conductivity. The flow of electrical current in

soils/rocks is mainly controlled by ionic charge transport taking place through water-filled pores, and by surface conduction mechanisms at mineral-ice-water interfaces(especially in the case of clay-rich materials, e.g. Revil and Glover, 1998).Hence, spatial and temporal resistivity variations in the subsurface depend on porosity, pore connectivity, mineralogy, water chemistry, water content, water saturation and temperature (e.g. Duvillard et al., 2018; Hermans et al., 2014; Ward et al., 2010).

We also added the following sentence at the end of the above paragraph:

According to Duvillard et al. (2018), the surface conduction dominates at low salinity instead of the electrolytic conduction, also supporting the water content control on electrical resistivity.

Section RESULTS:

Old version ( P13, L13-14):

[…] we find a linear relationship between temperature and resistivity. Temperature-resistivity linearity below the freezing point was also shown by laboratory experiments on bedrock samples (Krautblatter et al., 2010).

New version:

[…] we find a quasi-linear relationship between temperature and resistivity from ~0 °C down to our lowest temperature record (-3 °C). Within this temperature range, surface conduction is influenced by the increasing interface area; whereas in the case of ionic conduction the measured resistivity will be influenced by the decreasing amount of liquid water content, the decreasing mobility of the ions within the pore liquid as well as by the increasing salinity in the liquid phase (Oldenborger & LeBlanc, 2018). In the case of our network sites, the increase of surface conduction is compensated by a rather large decrease in saturation, which explains the resistivity increase.However, the theoretical exponential temperature dependence of the electric current (e.g. Robertson and MacDonald, 1962; Coperey et al., 2019) is not detectable in our field data, probably because of the narrow temperature range. Similarly, also Krautblatter et al. (2010) observed temperature-resistivity quasi-linearity by laboratory experiments on bedrock samples for sub-zero temperatures between -3 and 0 °C.

2. ERTM (i.e. monitoring) is only superior to ERT if the time is properly accounted for in the inversion e.g., through sequential inversion or real .5D+time (or 4D° inversion i.e.g., including reguilarization over time as it is done over space (see for instance Karaoulis M., A. Revil, D.D. Werkema, B. Minsley, W.F. Woodruff, and A. Kemna, Time-lapse 3D inversion of complex conductivity data using an active time constrained (ATC) approach, Geophysical Journal International, 187, 237–251, doi: 10.1111/j.1365-246X.2011.05156.x, 2011; Karaoulis M., A. Revil, D.D., Werkema, P. Tsourlos, and B.J. Minsley, IP4DI: A software for time-lapse 2D/3D DC-resistivity and induced polarization tomography, Computers & Geosciences, 54, 164-170, 2013.). Such a discussion is required, inverting time lapse dataset without the inclusion of time in the inversion can lead to serious errors due to noisy data and artefacts in the inversion process itself (do inversion at two different iteration at two different times will produced many spurious noamalies). The ATC approach can handle strong change in the data space so the argument "This is because our measurements are characterized by irregular time gaps and by strong spatial and temporal resistivity contrasts (active layer frozen/unfrozen)" is, in my opinion, incorrect.

We agree with the argumentation of the reviewer regarding the potential improvement of 4D inversion approaches for the inversion of monitoring datasets. The common geological background at each site (as geology is not expected to change over our time-scales) would serve as constraint in the inversion of the different (temporally linked) data sets. The rationale of our (simplified) approach was based on its compatibility and consistency with previous ERTM studies on permafrost, especially in an operational permafrost monitoring context.

The objective of our study is to present the consistent correlation observed between general geophysical parameters (i.e., electrical resistivity) and temperature for a monitoring network covering a broad range of

geological and geomorphological features. Our study aims at presenting in detail the characteristics of the network (including six sites with different instrumentation), as well as the available dataset and the inversion results. Within this permafrost monitoring context, we do not aim at extracting quantitative information based on the electrical resistivity images on the point scale. This would be (and has been) the aim of several case studies, where a wealth of additional data would come into play. A detailed analysis of the raw data has already been presented before (see the studies of Hilbich et al., 2008b, 2009, 2011; Hauck et al. 2002; Morard, 2011; Pogliotti et al., 2015; Rosset et al., 2013). Here, we are interested in the general trend and the comparison of such general responses over different sites. We believe therefore that the implementation and discussion of 4D inversion strategies are beyond the scope of the present study.

Hence, in our study we did not present selected pixels, but we opted to present a single value for each site/time corresponding to the average resistivity value (c.f. Figure 10). In that respect, the values represent the mean values of the selected permafrost layer in the inverted electrical model. We believe that this approach minimizes the risk of over-interpreting small-scale features in the electrical images.

In order to support our claim that the observed trend in the electrical properties is actually consistent for different study areas (i.e., geological and geomorphological features) and atmospheric conditions (as observed for the long monitoring period presented in our study) and independent of the inversion, we modified Figure 10. In this revised version of Figure 10, we present now also the temporal trend of the apparent resistivity data. Again, we do not analyse the temporal variations for a given pixel or quadrupole, but the average values of the permafrost layer, which minimizes the influence of random-error in the readings and uncertainties in the inversion. Both sub-plots Fig. 10a,b show a clear trend of the electrical parameters which is consistent with temperature.

Based on the above, we added the following paragraphs at the beginning of the discussion section 5.1:

> Inverting monitoring datasets without an explicit time-constraint in the inversion can produce errors due to noisy data (e.g., Karaoulis et al., 2013; Lesparre et al., 2017; Loke et al., 2014). The objectives of our study do not (yet) include the development and validation of an optimal inversion algorithm for an individual case study, but rather a simple and robust inversion approach that can detect the large-scale climate change signal (permafrost thawing) in the ERT data for very different field sites and conditions.
> […]
> In this case of continuous monitoring data sets at high temporal resolution, specific time-lapse inversion schemes may likely improve the tomograms significantly (as demonstrated by e.g. Doetsch et al., 2015). In our study, we excluded the interpretation of small-scale anomalies to reduce the uncertainty for the individual inversions.
> […]
> Moreover, both apparent resistivities and inverted resistivities (Fig. 10a,b) show similar decreasing trends supporting the assumption that existing inversion artefacts are smaller than the measured long-term trend.

As mentioned above, we modified Fig. 10 to include the apparent resistivity trend. Hereby, a large average over the permafrost domain is taken to reduce the effect of measurement and small-scale inversion errors. Note also, that we validated the inversion results with available active layer and temperature data.

[Figure]

Figure 1 Time series of the end-of-summer permafrost selected zone of (a) spatially averaged apparent resistivity (b) spatially averaged resistivity logarithm and (c) logarithm resistivity changes relative to the median of the end-of-summer time series. End-of-summer represents the second half of August, with extreme dates ranging between mid-July and mid-September for years without end-of-August data set. The grey dashed lines highlight the period (~2009-2013) of remarkable resistivity decrease at all sites.

We modified also the following paragraphs in the Discussion:

Old version (P15, L19-20)

Inverted tomograms may therefore have different resolutions, which may lead to the occurrence of artefacts in a time-lapse analysis.

New version:

Inverted tomograms may therefore have different resolutions, which may lead to the occurrence of artefacts in a time-lapse approach computing temporal differences, especially considering the low time continuity of our data sets (gaps in time) and the low amount of readings of a Wenner configuration.

Old version (P15, L24-26)

However, for detailed process studies on shorter time-scales, such as in Hilbich et al. (2011) for SCH, the same number of quadrupoles should be used for inversion to avoid inconsistencies and the occurrence of artefacts.

New version:

However, for detailed process studies on shorter time-scales or at high temporal resolution, such as presented in Hilbich et al. (2011) for SCH, the same number of quadrupoles should be used for inversion to avoid inconsistencies and the occurrence of artefacts. In this case of continuous monitoring data sets at high temporal resolution, specific time-lapse inversion schemes may likely improve the tomograms significantly (as demonstrated by e.g. Doetsch et al., 2015).

We also added the following paragraph in the Conclusion:

Following previous studies in ERT monitoring on mountain permafrost, we used independent inversions instead of time-lapse regularization of difference-inversions (e.g. Karaoulis et al., 2013, Lesparre et al., 2017; Loke et al., 2014) to provide a general overview of the long-term trend in the electrical parameters. The application and subsequent validation of a more advanced (e.g. 4D) inversion strategy (e.g. Doetsch et al., 2015) is beyond the scope of the present paper. In a forthcoming study, such an approach could first be applied at SCH, where resistivity data at high temporal resolution is available (Hilbich et al. 2011). This data availability is, however, the exception at operational permafrost monitoring stations, which currently favours a simple and robust data processing approach that is feasible for operational monitoring networks.

3. Personally I never accepted that reciprocity is necessarily the best test for error quantification and outliers removal. But I am fine with that. In principle, real time lapse inversion with the ATC technique filters out the data that are not correlated over time.

Reciprocity is a widely applied method that has demonstrated to be valid for a large number of studies at the field scale investigating different processes. (e.g., Slater et al., 2000; Koestel et al., 2008; Krautblatter et al., 2010; Flores Orozco et al., 2012). The advantage of such method is that it is sensitive to actual variations in the signal-to-noise ratio both in space and time. Moreover, it permits an improved identification of systematic errors in monitoring applications. Other studies have also demonstrated that time-lapse differences or time-regularization approaches can still result in the loss of resolution due to smoothing temporal variations in the data. But we understand the concerns of the reviewer and in our revised version of the manuscript we also mention that future research needs to include the analysis of the data using modern inversion algorithms. However, this discussion goes beyond the scope of this paper.

Old version (P8, L21-24):

New version:

> In view of the huge amount of data sets, automatic filtering procedures are essential to ensure consistent data processing. Several approaches with different levels of complexity exist. A time-lapse inversion scheme with time regularization (see e.g. Karaoulis et al., 2013; Lesparre et al., 2017; and the discussion section below) should in principle filter out the data, which are not correlated over time. The comparison of normal and reciprocal measurements (e.g. Flores-Orozco et al., 2012, 2018) is another common method to quantify data uncertainty.

4. RES2DINV and the L1 option is a very bad choice for inversion. The algorithm tends to produce step-like structrures when the number of iteration in the inversion process goes being 3. These are inversion artefacts. So this is bit sad that such an amazing dataset is inverted in a careless way. In addition, RES2DINV does not take into account the data covariance matrix. A big drawback when dealing with noisy data.

We thank the reviewer for the comment. We are aware that using too many iterations has to be avoided to not over-fit the data (i.e. introducing inversion artifacts, see also Hauck et al. 2003). This is why we choose to restrict the inversion to the 3$^{rd}$ iteration, as a compromise between keeping a good resolution and not over-fitting the data. Regarding the inversion scheme: over the past years, we did use different inversion schemes to invert our data, including e.g. CRTOMO (by Kemna, 2000), BERT/PyGIMLI (by Rücker et al., 2017), and did comparisons between the obtained inversion results obtained using different algorithms. However, we did not observe significant differences between the inversion results, especially not regarding their interpretation, which may also be a consequence of the Wenner configuration used for data collection (low number of quadrupoles). Similar results on comparisons between different inversion schemes have already been presented before by Hellman et al. (2016), and thus, we do not repeat it again in our study.

We rewrote the Inversion section (METHODS section 3.3) including the following modifications:

Old version (P9, L28):

> The inversion itself is conducted iteratively using the software RES2DINV (Loke, 2018) in batch mode. Similar to e.g. Caterina et al. (2017), Hilbich et al. (2011), Oldenborger and LeBlanc (2018), and Supper et al. (2014), we invert our data sets individually instead of using a time-lapse algorithm (designed to reduce systematic errors for time series with short time gaps, e.g. hours or days). This is because our measurements are characterized by irregular time gaps and by strong spatial and temporal resistivity contrasts (active layer frozen/unfrozen). We use the robust inversion scheme (L1-norm) with the so-called extended model to minimize the edge effects.

New version:

> We inverted our data using different inversion schemes and compared the obtained results (similarly to Hellman et al., 2016) and could not detect any significant differences between them that would be relevant for interpretation in a long-term permafrost context. Previous studies have also revealed that consistent temporal resistivity changes could be resolved using independent inversions of the data (e.g. Caterina et al., 2017; Hilbich et al., 2011; Oldenborger and LeBlanc, 2018; and Supper et al., 2014), instead of using a time-lapse algorithm (designed to reduce systematic errors for time series with short time gaps, e.g. hours or days). In this paper, in accordance with previous ERT studies at our field sites, the inversion itself is conducted iteratively using the software RES2DINV (Loke, 2018) in batch mode. We use the robust inversion scheme (L1-norm) with the so-called extended model to minimize the edge effects.
> More advanced time-lapse inversion schemes (as proposed e.g. by Karaoulis et al., 2013; Lesparre et al., 2017; Loke et al., 2014) can reduce the occurrence of inversion artefacts which may origin from the under-determined parts of the inversion problems of each individual tomogram. Difference plots of individual tomograms without explicit time-constraints may therefore contain artefacts, especially in regions with low sensitivity. In the following, small-scale anomalies in the resulting tomograms are therefore excluded from interpretation.

5. Section 4.1. what about induced polarization? Self-potential?

We did not completely understand the comment of the reviewer. Both induced polarization (IP) and self-potential (SP) are of course electrical methods that can be (and are!) used in the context of permafrost studies, but they are not part of the present study. We have currently ongoing IP and SP measurements at some of the field sites, but they are not on the same long-term monitoring time-scale and have also not been conducted at all of the field sites of the network. So we do not see the link between these methods and our results section in 4.1 as mentioned by the reviewer. We however mentioned those methods as future areas of research in the conclusion, which we renamed into "Conclusion and Outlook":

> Further electrical methods such as (spectral) induced polarization and self potential (e.g. Slater, 2007) may help to overcome limitations of the ERT method. However, applications of these methods to permafrost are still rare (e.g. Doetsch et al., 2015; Duvillard et al., 2018, Grimm and Stillman, 2015) and no long-term monitoring data sets exist yet

6. The approach for looking at time lapse change in resistivity is very dangerous because the resistivity data have not been inverted with appropriate time-lapse regularization techniques. The relevant literature on the subject is even not cited and it is has been shown in the literature that doing like this brings a lot of artefacts in the computed changes because of inversion artefacts.

See response above to reviewer point 2.

7. Section 4.3 is far from our level of knowledge regarding the underlying physics of the relationship between resistivity and temperature, which is non-linear, see Duvoiillard etal. 2018 and Coperey et al. 2019 (both in JGR-SE).

See response above to reviewer point 1.

**Short comment #2: Review from Sara Bazin (June 4[th] 2019)**

This manuscript reports results of 2D ERT monitoring surveys at six sites in the western Alps. The authors draw conclusions on the state and the degradation of permafrost likely related to climate changes, which deserves publication in The Cryosphere.

This paper is very well written and the figures are well presented. The abstract is concise and the introduction gives a good overview and presents the motivation. This long-term dataset is extremely valuable. The data quality is variable and the data selection is well explained. The data inversion seems well done. The way to illustrate the seasonal effect in Figures 4 and 8 is very nice. Figure 10 is a very clever and innovative way to illustrate the long-term variation.

The amount of work to compute these tomographic images is probably enormous and the results are impressive. The analysis is thorough and the interpretation is very interesting. Thus, I recommend publication and I have very few comments to improve this manuscript.

We are thankful to the reviewer for this positive feedback.

The only requests/suggestions would be:
- Add the reciprocal measurements in the supplement. It is a bit useless to talk about them, without showing figures.

Here, we were apparently not clear enough. We mentioned the reciprocal measurements while discussing commonly used filtering techniques. However, the long-term data acquisition in this study (see section 3.2.2) does not include reciprocal measurements, for a gain of time and energy (all our instrumentations rely on batteries).

- Check the format for referencing several publication in the text. I understand that the rule is chronologically before alphabetically.

From the Cryosphere manuscript preparation guidelines for authors, one can read that the rule is different for the reference list and the in-text citation. The reference list rule is very detailed, whereas the in-text citation is up to the authors: "In terms of in-text citations, the order can be based on relevance, as well as chronological or alphabetical listing, depending on the author's preference." from https://www.the-cryosphere.net/for_authors/manuscript_preparation.html (June, 4$^{th}$, 2019). We however, harmonized our in-text citation order alphabetically, if not based on relevance.

- Add a more general paragraph on ERTM time series. I feel that some references on resistivity monitoring are missing.

Following the reviewer advice, we added references to other ERTM time series studies outside from the cryosphere application in the introduction.

Section INTRODUCTION:

> Electrical resisitivity tomography (ERT) and its monitoring variant ERTM has been used for many years in other geoscientific fields such as landslide monitoring, irrigation/infiltration studies and landfill monitoring (e.g., Barker and Moore, 1998: LaBrecque and Yang, 2001; Loke, 1999; Suzuki and Higashi, 2001).

Comments:
Page 2 Line 2: Change "may" by "will"
Page 4 – line 4: ALT is already explained earlier
Page 4 line 18: explain the short and long-wave radiations for those who are not familiar with meteorological stations.
Page 6, line 28: I would remove "Here,..."
Page 9, line 22. Please add the user-defined site-specific thresholds for the inversions in table 1.

Changed accordingly

Page 9: I would rephrase or expand the sentence in line 25.

We modified our manuscript according to the reviewer suggestions.
Old version (P9, L24-25):

> However, such cases would already be excluded by step 2, which is therefore essential to keep a high tomogram resolution.

New version:

> However, such cases (high filtering and low inversion error) would already be excluded by step 2, which is therefore essential to keep a high tomogram resolution.

Page10: I would also rephrase or expand the sentence in lines 9-12.

Old version (P10, L9-11):

> As a standardised and non-empirical histogram analysis can indeed not be conducted with such a low number of bins (cf. Fig. S3), these kind of manually defined thresholds based on expert knowledge have also been used in similar studies (cf. Supper et al., 2014).

New version:

> As a standardised and non-empirical histogram analysis can indeed not be conducted with such a low number of bins (statistically, the number of measurements is low cf. Fig. S3), manually defined thresholds based on expert knowledge were used, similarly to other studies (cf. Supper et al., 2014).

Figure 2: I would not use blue for ALT1 and ALT2. It is confusing with the 2 other blues (LAGT10m 1 and 2). In the caption, add "Where available, data are given for two boreholes (1 and 2).

We modified Figure 2 according to the reviewer suggestion.

Figure 10: I really like these 2 figures. The right one is however a bit difficult to understand/follow. The yellow background is not a good choice. Maybe you could use thicker and coloured lines in this area.

We thank the reviewer. We removed the yellow background and thought that the existing thicker grey lines are enough to highlight the remarkable period. We slightly modified the legend accordingly.

**Additional references (References already present in the first version of the manuscript are not repeated here):**

Barker, R. and Moore J.: The application of time-lapse electrical tomography in groundwater studies. Geophysics. 17. 10.1190/1.1437878, 1998.

Coperey, A., Revil, A., Abdulsamad, F., Stutz, B., Duvillard, P. A., and Ravanel, L.: Low frequency induced polarization of porous media undergoing freezing: preliminary observations and modeling. J. Geophys. Res. - Sol. Ea. 124, doi:10.1029/2018JB017015, 2019.

Dafflon, B., Oktem, R., Peterson, J., Ulrich, C., Tran, A. P., Romanovsky, V., and Hubbard, S. S.: Coincident aboveground and belowground autonomous monitoring to quantify covariability in permafrost, soil, and vegetation properties in Arctic tundra, J. Geophys. Res. Biogeosci., 122, 1321– 1342, doi:10.1002/2016JG003724, 2017.

Duvillard, P. A., Revil, A., Qi, Y., Soueid Ahmed, A., Coperey, A., and Ravanel, L.: Three-dimensional electrical conductivity and induced polarization tomography of a rock glacier, J. Geoph. Res. 123, doi:10.1029/2018JB015965, 2018.

Grimm, R. E. and Stillman, D. E.: Field test of detection and characterization of subsurface ice using broadband spectral induced polarization. Permafrost Periglac., 26, 28–38, doi: 10.3189/2015jog15j113, 2015.

Hellman, K., Johansson, S., Olsson, P., and Dahlin, T.: Resistivity Inversion Software Comparison. Near Surface Geoscience Conference 2016, doi:10.3997/2214-4609.201602016, 2016.

Karaoulis, M. , Tsourlos, P. , Kim, J. and Revil, A.: 4D time-lapse ERT inversion: introducing combined time and space constraints. Near Surf. Geophys., 12: 25-34. doi:10.3997/1873-0604.2013004, 2013.

Kellerer-Pirklbauer, A. and Kaufmann, V.: About the relationship between rock glacier velocity and climate parameters in central Austria. Austrian J. Earth Sc. 105. 94-112, 2012.

Klein, A. K. and Carlos Santamarina, J.: Electrical Conductivity in Soils: Underlying Phenomena. Journal of Environmental and Engineering Geophysics – J. Environ. Eng. Geophys. 8. 10.4133/JEEG8.4.263, 2003.

Koestel, J., Kemna, A., Javaux, M., Binley, A., and Vereecken H.: Quantitative imaging of solute transport in an unsaturated and undisturbed soil monolith with 3-D ERT and TDR,Water Resour. Res.,44,W12411, doi:10.1029/2007WR006755, 2008.

LaBrecque D. J. and Yang X.: Difference inversion of ERT data: A fast inversion method for 3-D in situ monitoring. J. Environ. Eng. Geophys. 1083-1363, 6, 83–8, 2001.

LaBrecque, D., Miletto, M., Daily, W., Ramirez, A., and Owen, E.: The effects of 'Occam' inversion of resistivity tomography data. Geophysics. 61. 538-548. 10.1190/1.1443980, 1996.

Lesparre, N, Nguyen, F., Kemna, A., Robert, T., Hermans, T., Daoudi, M., and Flores Orozco, A.: A new approach for time-lapse data weighting in electrical resistivity tomography. Geophysics, 82(6), E325–E333, 2017.

Loke M. H.: Time-lapse resistivity imaging inversion: Environmental and Engineering Geophysical Society European Section, Meeting, Proceedings, Em1, doi:10.4133/1.2922877, 1999.

Loke, M. ,Dahlin, T. and Rucker, D. F.: Smoothness-constrained time-lapse inversion of data from 3D resistivity surveys. Near Surf. Geophys., 12: 5-24. doi:10.3997/1873-0604.2013025, 2014.

Revil, A. and Glover, P. W. J.: Nature of surface electrical conductivity in natural sands, sandstones, and clays. Geophys. Res. Lett., doi:10.1029/98GL00296, 1998.

Robertson E. I. and MacDonald W. J. P.: Electrical resistivity and ground temperature at Scott Base, Antarctica, New Zeal. J. Geol. Geop., 5:5, 797-809, doi:10.1080/00288306.1962.10417639, 1962

Rücker, C., Günther, T., and Wagner, F. M.: pyGIMLi: An open-source library for modelling and inversion in geophysics, Comput. Geosci., 109, 106–123, doi:10.1016/j.cageo.2017.07.011, 2017.

Slater, L.: Near Surface Electrical Characterization of Hydraulic Conductivity: From Petrophysical Properties to Aquifer Geometries—A Review. Surv. Geophys. 28. 169-197, doi:10.1007/s10712-007-9022-y, 2007.

Slater, L., Binley, A., Daily, W., and Johnson, R.: Cross-hole electrical imaging of a controlled saline tracer injection. J. Appl. Geophys. 44, 85–102, 2000.

[revised manuscript text omitted]

---

## Author Response (AR2)

Dear editor,

Thank you very much for your careful review, scientific, language and grammatical corrections, which indeed improve clarity and readability.

The manuscript was corrected accordingly and checked throughout once more.

We hope that we have properly address your concerns.

Sincerely,

Coline Mollaret on behalf of all authors.

**Point-by-point responses**

I leave it to you to make the decision, but most commonly the dimension is referred to as "metre" and a measurement device is a "…meter"

Thank you for pointing it out.

a chimney effect would be one way. Balch (1900) describes the reversible air circulation.

Thank you for pointing out this reference that I was not aware of. I only had time to read partly the book of Balch (1900). However, from other references referring to Blach (1900), i.e. Wicky and Hauck (2017), Schneider et al. (2012), Delaloye and Lambiel (2015), I understand that Balch-effect refers to convection effect only, and refers to flat ground, that is why I preferred to keep the term "chimney effect", which is described as being reversible and occurring in coarse debris terrain like talus slopes or rock glaciers.

Hysteresis effects in freezing-thawing soils in relation to the influence of soil characteristics on water content are well known and should be cited here. A relatively recent paper is H. Tian, C. Wei, H. Wei, J. Zhou, Freezing and thawing characteristics of frozen soils: bound water content and hysteresis phenomenon, Cold Reg. Sci. Technol., 103 (2014), pp. 74-81, but there are other older touchstone papers as well.

Thank you for this reference of Tian et al. (2014) that I was not aware of.

[revised manuscript text omitted]

---

## Author Response (AR3)

Dear Peter Morse,

We sincerely thank you for the further revisions that you suggested to improve our manuscript.

We took all of them into account, except the change of "Fig." into "Figure" when it appeared in running text, as it is preconized here: https://www.the-cryosphere.net/for_authors/manuscript_preparation.html, where one can read:

*The abbreviation "Fig." should be used when it appears in running text and should be followed by a number unless it comes at the beginning of a sentence, e.g.: "The results are depicted in Fig. 5. Figure 9 reveals that...".*

Best regards,

Coline Mollaret

On behalf of all co-authors

[revised manuscript text omitted]